# SPARSE-PGD: AN EFFECTIVE AND EFFICIENT ATTACK FOR $l_0$ BOUNDED ADVERSARIAL PERTURBATION

## ABSTRACT

This work focuses on sparse adversarial perturbations bounded by $l_0$ norm. We propose a white-box PGD-like attack method named sparse-PGD to effectively and efficiently generate such perturbations. Furthermore, we combine sparse-PGD with a black-box attack to comprehensively and more reliably evaluate the models' robustness against $l_0$ bounded adversarial perturbations. Moreover, due to the efficiency of sparse-PGD, we explore utilizing it to conduct adversarial training to build robust models against sparse perturbations. Extensive experiments demonstrate that our proposed attack algorithm can achieve better performance than baselines. Our adversarially trained model also shows the strongest robustness against various sparse attacks.

## 1 INTRODUCTION

Deep learning has been developing tremendously fast in the last decade. However, it is shown vulnerable to adversarial attacks: imperceivable adversarial perturbations (Szegedy et al., 2013; Kurakin et al., 2016) could change the prediction of a classifier without altering the semantic content of the input, which poses great challenges in safety-critical systems. Among different kinds of adversarial perturbations, the ones bounded by $l_\infty$ or $l_2$ norms are mostly well-studied (Goodfellow et al., 2014; Madry et al., 2017; Zhang et al., 2019c) and benchmarked (Croce et al., 2020), because the adversarial budgets, i.e., the sets of all allowable perturbations, are convex, which facilities theoretical analyses and algorithm design. By contrast, we focus on perturbations bounded by $l_0$ norm in this work. These perturbations are sparse and quite common in physical scenarios, including broken pixels in LED screens to fool object detection models and adversarial stickers on road signs to make an auto-driving system fail (Papernot et al., 2017; Akhtar & Mian, 2018; Xu et al., 2019).

However, constructing $l_0$ bounded adversarial perturbations is challenging as the corresponding adversarial budget is non-convex. Therefore, it is difficult to apply gradient-based methods, such as projected gradient descent (PGD) (Madry et al., 2017) to obtain a strong adversarial perturbation efficiently. Existing methods to generate sparse perturbations Modas et al. (2018); Croce & Hein (2019c); Su et al. (2019); Dong et al. (2020); Croce et al. (2022) either cannot control the $l_0$ norm of perturbations or have prohibitively high computational complexity, which makes them inapplicable for adversarial training to obtain robust models against sparse perturbations. The perturbations bounded by $l_1$ norm are the closest scenario to sparse perturbations among convex adversarial budgets defined by an $l_p$ norm. Nevertheless, adversarial training in this case (Tramer & Boneh, 2019; Croce & Hein, 2021) still suffers from issues such as slow convergence and instability. Jiang et al. (2023) demonstrates that these issues arise from non-sparse perturbations bounded by $l_1$ norm. In other words, $l_1$ adversarial budget still cannot guarantee the sparsity of the perturbations. Thus, it is necessary to study the case of $l_0$ bounded perturbations.

In this work, we propose a white-box attack named sparse-PGD (sPGD) to generate sparse perturbations bounded by $l_0$ norm. Specifically, we decompose the sparse perturbation $\boldsymbol{\delta}$ as the product of a magnitude tensor $\boldsymbol{p}$ and a binary sparse mask $\boldsymbol{m}$: $\boldsymbol{\delta} = \boldsymbol{p} \odot \boldsymbol{m}$, where $\boldsymbol{p}$ and $\boldsymbol{m}$ determine the magnitudes and the locations of perturbed features, respectively. We adopt PGD-like algorithms to update $\boldsymbol{p}$ and $\boldsymbol{m}$. However, it is challenging to directly optimize the binary mask $\boldsymbol{m}$ in the discrete space. We thereby introduce an alternative continuous variable $\widetilde{\boldsymbol{m}}$ to approximate $\boldsymbol{m}$ and update $\widetilde{\boldsymbol{m}}$ by gradient-based methods, $\widetilde{\boldsymbol{m}}$ is then transformed to $\boldsymbol{m}$ by projection to the discrete space. Due to the sparsity of $\boldsymbol{m}$ by the projection operator, the gradient of $\boldsymbol{p}$ is sparse, which may lead

to slow convergence by coordinate descent. Therefore, we remove the projection operator in the backpropagation to obtain the unprojected gradient of $\boldsymbol{p}$. We use both the original sparse gradient and the unprojected gradient of $\boldsymbol{p}$ to boost the attack performance. Moreover, we design a random reinitialization mechanism to enhance the exploration capability for the mask $\boldsymbol{m}$. On top of sPGD, we propose sparse-AutoAttack (sAA), which is the ensemble of the white-box sPGD and another black-box sparse attack, for a more comprehensive and reliable evaluation against $l_0$ bounded perturbations. Through extensive experiments, we show that our method achieves better performance than other sparse attacks.

We also explore adversarial training against sparse attacks. In this context, the attack method will be called in each mini-batch update, so it should be both effective and efficient. Compared with existing sparse attack methods, our proposed sPGD performs much better when using a small number of iterations, making it feasible for adversarial training. Empirically, models adversarially trained by sPGD demonstrate the strongest robustness against various sparse attacks.

We summarize the contributions of this paper as follows:

1. We propose an effective and efficient attack algorithm called sparse-PGD (sPGD) to generate $l_0$ bounded adversarial perturbation. sPGD achieves significantly better performance than existing methods when using limited iterations, which makes it feasible for adversarial training to obtain robust models against sparse perturbations.

2. We propose an ensemble of sparse attacks called sparse-AutoAttack (sAA) for reliable robustness evaluation against $l_0$ bounded perturbation.

3. We conduct extensive experiments to demonstrate that our attack methods achieve impressive performance in terms of both effectiveness and efficiency. Based on sPGD, we conduct adversarial training against $l_0$ bounded perturbations. Models adversarially trained by our attack method show the strongest robustness against various sparse attacks.

## 2 PRELIMINARIES

We use image classification as an example, although the methods proposed in this work are applicable to any classification model. Under $l_p$ bounded perturbations, the robust learning aims to solve the following min-max optimization problem.

$$\min_{\boldsymbol{\theta}} \frac{1}{N} \sum_{i=1}^{N} \max_{\boldsymbol{\delta}_i} \mathcal{L}(\boldsymbol{\theta}, \boldsymbol{x}_i + \boldsymbol{\delta}_i), \quad \text{s.t. } ||\boldsymbol{\delta}_i||_p \leq \epsilon, \ 0 \leq \boldsymbol{x}_i + \boldsymbol{\delta}_i \leq 1. \tag{1}$$

where $\boldsymbol{\theta}$ denotes the parameters of the model and $\mathcal{L}$ is the loss objective function. $\boldsymbol{x}_i \in \mathbb{R}^{h \times w \times c}$ is the input image where $h$, $w$ and $c$ represent the height, width, and number of channels, respectively. $\boldsymbol{\delta}_i$ has the same shape as $\boldsymbol{x}_i$ and represents the perturbation. The perturbations are constrained by its $l_p$ norm and the bounding box. In this regard, we use the term *adversarial budget* to represent the set of all allowable perturbations. Adversarial attacks focus on the inner maximization problem of (1) and aim to find the optimal adversarial perturbation, while adversarial training focuses on the outer minimization problem of (1) and aims to find a robust model parameterized by $\boldsymbol{\theta}$. Due to the high dimensionality and non-convexity of the loss function when training a deep neural network, Weng et al. (2018) has proven that solving the problem (1) is at least NP-complete.

In this work, we study the $l_0$ bounded perturbations. For image inputs, we consider the pixel sparsity, which is more meaningful than feature sparsity and consistent with existing works (Croce & Hein, 2019c; Croce et al., 2022). That is, a pixel is considered perturbed if *any* of its channel is perturbed, and sparse perturbation means few pixels are perturbed.

## 3 RELATED WORKS

**Non-Sparse Attacks:** The pioneering work Szegedy et al. (2013) finds the adversarial perturbations to fool image classifiers and proposes a method to minimize the $l_2$ norm of such perturbations. To more efficiently generate adversarial perturbations, the fast gradient sign method (FGSM) Goodfellow et al. (2014) generates $l_\infty$ perturbation in one step, but its performance is significantly surpassed

by the multi-step variants Kurakin et al. (2017). Projected Gradient Descent (PGD) (Madry et al., 2017) further boosts the attack performance by using iterative updating and random initialization. Specifically, each iteration of PGD updates the adversarial perturbation $\boldsymbol{\delta}$ by:

$$\boldsymbol{\delta} \longleftarrow \Pi_{\mathcal{S}}(\boldsymbol{\delta} + \alpha \cdot s(\nabla_{\boldsymbol{\delta}}\mathcal{L}(\boldsymbol{\theta}, \boldsymbol{x} + \boldsymbol{\delta}))) \tag{2}$$

where $\mathcal{S}$ is the adversarial budget, $\alpha$ is the step size, $s : \mathbb{R}^{h \times w \times c} \to \mathbb{R}^{h \times w \times c}$ selects the steepest ascent direction based on the gradient of the loss $\mathcal{L}$ with respect to the perturbation. Inspired by the first-order Taylor expansion, (Madry et al., 2017) derives the steepest ascent direction for $l_2$ bounded and $l_\infty$ bounded perturbations to efficiently find strong adversarial examples; SLIDE (Tramer & Boneh, 2019) and $l_1$-APGD (Croce & Hein, 2021) use $k$-coordinate ascent to construct $l_1$ bounded perturbations, which is shown to suffer from the slow convergence (Jiang et al., 2023).

Besides the attacks that have access to the gradient of the input (i.e., white-box attacks), there are also black-box attacks that do not have access to model parameters, including the ones based on gradient estimation through finite differences (Bhagoji et al., 2018; Ilyas et al., 2018a;b; Tu et al., 2018; Uesato et al., 2018) and the ones based on evolutionary strategies or random search (Alzantot et al., 2018; Guo et al., 2019). To improve the query efficiency of black-box attacks, Al-Dujaili & OReilly (2019); Moon et al. (2019); Meunier et al. (2019); Andriushchenko et al. (2019) generate adversarial perturbation at the corners of the adversarial budget.

To more reliably evaluate the robustness, (Croce & Hein, 2020) proposes AutoAttack (AA) which consists of an ensemble of several attack methods, including both black-box and white-box attacks. (Croce & Hein, 2021) extends AA to the case of $l_1$ bounded perturbations and proposes AutoAttack-$l_1$ (AA-$l_1$). Although the $l_1$ bounded perturbations are usually sparse, Jiang et al. (2023) demonstrates that AA-$l_1$ is able to find non-sparse perturbations that cannot be found by SLIDE to fool the models. That is to say, $l_1$ bounded adversarial perturbations are not guaranteed to be sparse. We should study perturbations bounded by $l_0$ norm.

**Sparse Attacks:** For perturbations bounded by $l_0$ norm, directly adopting vanilla PGD as in Equation (2) leads to suboptimal performance due to the non-convexity nature of the adversarial budget: $\text{PGD}_0$ (Croce & Hein, 2019c), which updates the perturbation by gradient ascent and project it back to the adversarial budget, turns out very likely to trap in the local maxima. SparseFool (Modas et al., 2018) and GreedyFool (Dong et al., 2020) also generate sparse perturbations, but they do not strictly restrict the $l_0$ norm of perturbations. If we project their generated perturbations to the desired $l_0$ ball, their performance will drastically drop. Sparse Adversarial and Interpretable Attack Framework (SAIF) (Imtiaz et al., 2022) is similar to our method in that SAIF also decomposes the $l_0$ perturbation into a magnitude tensor and sparsity mask, but it uses the Frank-Wolfe algorithm (Frank et al., 1956) to separately update them. SAIF turns out to get trapped in local minima and shows poor performance on adversarially trained models. Besides white-box attacks, there are black-box attacks to generate sparse adversarial perturbations, including CornerSearch (Croce & Hein, 2019c) and Sparse-RS (Croce et al., 2022). However, these black-box attacks usually require thousands of queries to find an adversarial example, making it difficult to scale up to large datasets.

**Adversarial Training:** Despite the difficulty in obtaining robust deep neural network, adversarial training (Madry et al., 2017; Croce & Hein, 2019b; Sehwag et al., 2021; Rebuffi et al., 2021; Gowal et al., 2021; Rade & Moosavi-Dezfooli, 2021; Cui et al., 2023; Wang et al., 2023) stands out as a reliable and popular approach to do so (Athalye et al., 2018; Croce & Hein, 2020). It generates adversarial examples first and then uses them to optimize model parameters. Despite effective, adversarial training is time-consuming due to multi-step attacks. (Shafahi et al., 2019; Zhang et al., 2019a; Wong et al., 2020; Sriramanan et al., 2021) use weaker but faster one-step attacks to reduce the overhead, but they may suffer from catastrophic overfitting (Kang & Moosavi-Dezfooli, 2021): the model overfits to these weak attacks during training instead of achieving true robustness to various attacks. Kim et al. (2020); Andriushchenko & Flammarion (2020); Golgooni et al. (2021); de Jorge et al. (2022) try to overcome catastrophic overfitting while maintaining efficiency.

Compared with $l_\infty$ and $l_2$ bounded perturbations, adversarial training against $l_1$ bounded perturbations is shown to be even more time-consuming to achieve the optimal performance (Croce & Hein, 2021). In the case of sparse perturbations, there is currently no adversarial training method due to the large computational overhead to generate sparse perturbations. In this work, we propose an effective and efficient sparse attack that enables us to conduct adversarial training against sparse perturbations. The obtained model demonstrates the best robustness against various sparse attacks.

It should be noted that we only regard this as a baseline of adversarial training against $l_0$ bounded perturbations. Exploring methods to further boost performance will be the future work.

## 4 METHODS

In this section, we introduce sparse-PGD (sPGD): a white-box attack that generates sparse perturbations. Based on sPGD, we propose sparse-AutoAttack (sAA), an ensemble of various sparse attacks, for reliable robustness evaluation against $l_0$ bounded adversarial perturbation. In the end, we incorporate sPGD into adversarial training to boost the model's robustness against sparse perturbations.

### 4.1 SPARSE-PGD (SPGD)

Inspired by SAIF (Imtiaz et al., 2022), we decompose the sparse perturbation $\boldsymbol{\delta}$ into a magnitude tensor $\boldsymbol{p} \in \mathbb{R}^{h \times w \times c}$ and a sparsity mask $\boldsymbol{m} \in \{0, 1\}^{h \times w \times 1}$, i.e., $\boldsymbol{\delta} = \boldsymbol{p} \odot \boldsymbol{m}$. Therefore, the loss objective function the attacker aims to maximize can be rewritten as

$$\max_{\|\boldsymbol{\delta}\|_0 \leq k, 0 \leq \boldsymbol{x}+\boldsymbol{\delta} \leq 1} \mathcal{L}(\boldsymbol{\theta}, \boldsymbol{x} + \boldsymbol{\delta}) = \max_{\boldsymbol{p} \in \mathcal{S}_{\boldsymbol{p}}, \boldsymbol{m} \in \mathcal{S}_{\boldsymbol{m}}} \mathcal{L}(\boldsymbol{\theta}, \boldsymbol{x} + \boldsymbol{p} \odot \boldsymbol{m}). \tag{3}$$

The feasible sets for $\boldsymbol{p}$ and $\boldsymbol{m}$ are $\mathcal{S}_{\boldsymbol{p}} = \{\boldsymbol{p} \in \mathbb{R}^{h \times w \times c} | 0 \leq \boldsymbol{x} + \boldsymbol{p} \leq 1\}$ and $\mathcal{S}_{\boldsymbol{m}} = \{\boldsymbol{m} \in \{0, 1\}^{h \times w \times 1} | \|\boldsymbol{m}\|_0 \leq k\}$, respectively. sPGD iteratively updates $\boldsymbol{p}$ and $\boldsymbol{m}$ until finding a successful adversarial example or reaching the maximum iteration number. We elaborate on the details below.

**Update Magnitude Tensor $\boldsymbol{p}$:** The magnitude tensor $\boldsymbol{p}$ is only constrained by the input domain. In the case of RGB images, the input is bounded by 0 and 1. Note that the constraints on $\boldsymbol{p}$ are elementwise and similar to those of $l_\infty$ bounded perturbations. Therefore, instead of greedy or random search (Croce & Hein, 2019c; Croce et al., 2022), we utilize PGD in the $l_\infty$ case, i.e., use the sign of the gradients, to optimize $\boldsymbol{p}$ as demonstrated by Eq. (4), with $\alpha$ being the step size.

$$\boldsymbol{p} \longleftarrow \Pi_{\mathcal{S}_{\boldsymbol{p}}} \left( \boldsymbol{p} + \alpha \cdot \mathtt{sign}(\nabla_{\boldsymbol{p}} \mathcal{L}(\boldsymbol{\theta}, \boldsymbol{x} + \boldsymbol{p} \odot \boldsymbol{m})) \right), \tag{4}$$

**Update Sparsity Mask $\boldsymbol{m}$:** The sparsity mask $\boldsymbol{m}$ is binary and constrained by its $l_0$ norm. Directly optimizing the discrete variable $\boldsymbol{m}$ is challenging, so we update its continuous alternative $\widetilde{\boldsymbol{m}} \in \mathbb{R}^{h \times w \times 1}$ and project $\widetilde{\boldsymbol{m}}$ to the feasible set $\mathcal{S}_{\boldsymbol{m}}$ before multiplying it with the magnitude tensor $\boldsymbol{p}$ to calculate the sparse perturbation $\boldsymbol{\delta}$. Specifically, $\widetilde{\boldsymbol{m}}$ is updated by gradient ascent. Projecting $\widetilde{\boldsymbol{m}}$ to the feasible set $\mathcal{S}_{\boldsymbol{m}}$ is to set the $k$-largest elements in $\widetilde{\boldsymbol{m}}$ to 1 and the rest to 0. In addition, we adopt the sigmoid function to normalize the elements of $\widetilde{\boldsymbol{m}}$ before projection. Mathematically, the update rules for $\widetilde{\boldsymbol{m}}$ and $\boldsymbol{m}$ are demonstrated as follows:

$$\widetilde{\boldsymbol{m}} \longleftarrow \widetilde{\boldsymbol{m}} + \beta \cdot \nabla_{\widetilde{\boldsymbol{m}}} \mathcal{L} / (\|\nabla_{\widetilde{\boldsymbol{m}}} \mathcal{L}\|_2 + \gamma), \tag{5}$$

$$\boldsymbol{m} \longleftarrow \Pi_{\mathcal{S}_{\boldsymbol{m}}}(\sigma(\widetilde{\boldsymbol{m}})) \tag{6}$$

where $\beta$ is the step size for updating the sparsity mask's continuous alternative $\widetilde{\boldsymbol{m}}$, $\sigma(\cdot)$ denotes the sigmoid function and $\gamma$ is a small constant to avoid the denominator becoming zero. The gradient $\nabla_{\widetilde{\boldsymbol{m}}} \mathcal{L}$ is calculated at the point $\boldsymbol{\delta} = \boldsymbol{p} \odot \Pi_{\mathcal{S}_{\boldsymbol{m}}}(\sigma(\widetilde{\boldsymbol{m}}))$, where the loss function is not always differentiable. We demonstrate how to estimate the update direction in the next part. Furthermore, to prevent the magnitude of $\widetilde{\boldsymbol{m}}$ from becoming explosively large, we do not update $\widetilde{\boldsymbol{m}}$ when $\|\nabla_{\widetilde{\boldsymbol{m}}} \mathcal{L}\|_2 < 2 \cdot \gamma$, which indicates that $\widetilde{\boldsymbol{m}}$ is located in the saturation zone of sigmoid function.

**Backward Function:** Based on Eq. (3), we can calculate the gradient of the magnitude tensor $\boldsymbol{p}$: $\frac{\partial \mathcal{L}}{\partial \boldsymbol{p}} = \nabla_{\boldsymbol{\delta}} \mathcal{L}(\boldsymbol{\theta}, \boldsymbol{x} + \boldsymbol{\delta}) \odot \boldsymbol{m}$ and use $g_{\boldsymbol{p}}$ to represent this gradient for notation simplicity. There are at most $k$ non-zero elements in the mask $\boldsymbol{m}$, so $g_{\boldsymbol{p}}$ is sparse and also has at most $k$ non-zero elements. That is to say, we update at most $k$ elements of the magnitude tensor $\boldsymbol{p}$ based on the gradient $g_{\boldsymbol{p}}$. Just like coordinate descent, this may result in suboptimal performance since most elements of $\boldsymbol{p}$ are unchanged in each iterative update. To tackle this problem, we discard the projection to the binary set $\mathcal{S}_{\boldsymbol{m}}$ when calculating the gradient and use the *unprojected gradient* $\widetilde{g}_{\boldsymbol{p}}$ to update $\boldsymbol{p}$. Based on Eq. (6), we have $\widetilde{g}_{\boldsymbol{p}} = \nabla_{\boldsymbol{\delta}} \mathcal{L}(\boldsymbol{\theta}, \boldsymbol{x} + \boldsymbol{\delta}) \odot \sigma(\widetilde{\boldsymbol{m}})$. The idea of the unprojected gradient is inspired by training pruned neural networks and lottery ticket hypothesis (Frankle & Carbin, 2019; Ramanujan et al., 2020; Fu et al., 2021; Liu et al., 2022). All these methods train importance scores to prune the model parameters but update the importance scores based on the whole network instead of the pruned sub-network to prevent the sparse update, which leads to suboptimal performance.

In practice, the performance of using $g_{\boldsymbol{p}}$ and $\widetilde{g}_{\boldsymbol{p}}$ to optimize $\boldsymbol{p}$ is complementary. The sparse gradient $g_{\boldsymbol{p}}$ is consistent with the forward propagation and is thus better at exploitation. By contrast, the unprojected gradient $\widetilde{g}_{\boldsymbol{p}}$ updates the $\boldsymbol{p}$ by a dense matrix and is thus better at exploration. In view of this, we set up an ensemble of attacks with both gradients to balance exploration and exploitation.

When calculating the gradient of the continuous alternative $\widetilde{m}$, we have $\frac{\partial \mathcal{L}}{\partial \widetilde{m}} = \frac{\partial \mathcal{L}(\theta, \boldsymbol{x}+\boldsymbol{\delta})}{\partial \boldsymbol{\delta}} \odot \boldsymbol{p} \odot \frac{\partial \Pi_{\mathcal{S}_m}(\sigma(\widetilde{m}))}{\partial \widetilde{m}}$. Since the projection to the set $\mathcal{S}_m$ is not always differentiable, we discard the projection operator and use the approximation $\frac{\partial \Pi_{\mathcal{S}_m}(\sigma(\widetilde{m}))}{\partial \widetilde{m}} \simeq \sigma'(\widetilde{m})$ to calculate the gradient.

**Random Reinitialization:** Due to the projection to the set $\mathcal{S}_m$ in Eq. (6), the sparsity mask $\boldsymbol{m}$ changes only when the relative magnitude ordering of the continuous alternative $\widetilde{m}$ changes. In other words, slight changes in $\widetilde{m}$ usually mean no change in $\boldsymbol{m}$. As a result, it is quite easy for $\boldsymbol{m}$ to get trapped in a local maximum. To solve this problem, we propose a random reinitialization mechanism. Specifically, when the attack fails, i.e., the model still gives the correct prediction, and the current sparsity mask $\boldsymbol{m}$ remains unchanged for three consecutive iterations, the continuous alternative $\widetilde{m}$ will be randomly reinitialized for better exploration.

To summarize, we provide the pseudo-code of the untargetted version of sparse PGD (sPGD) in Algorithm 1. While the untargeted version aims to maximize the loss objective of the correct label, targeted sPGD minimizes the loss objective of the targeted label. When evaluating the top $C$ incorrect labels with the highest confidence scores in the targeted sPGD, its computational complexity becomes $C$ times that of the untargeted counterpart.

---

**Algorithm 1** Untargeted sPGD

> **Input:** Clean image: $\boldsymbol{x} \in [0,1]^{h \times w \times c}$; Model parameters: $\boldsymbol{\theta}$; Max iteration number: $T$; $l_0$ budget: $k$; Step size: $\alpha$, $\beta$; Small constant: $\gamma$

1: Random initialize $\boldsymbol{p}$ and $\widetilde{m}$
2: **for** $i = 0, 1, ..., T-1$ **do**
3:     $\boldsymbol{m} = \Pi_{\mathcal{S}_m}(\sigma(\widetilde{m}))$
4:     Calculate the loss $\mathcal{L}(\boldsymbol{\theta}, \boldsymbol{x}+\boldsymbol{p} \odot \boldsymbol{m})$
5:     $g_{\boldsymbol{p}} = \nabla_{\boldsymbol{\delta}}\mathcal{L} \odot \sigma(\widetilde{m})$ **if** unprojected **else** $\nabla_{\boldsymbol{\delta}}\mathcal{L} \odot \boldsymbol{m}$            $\triangleright \boldsymbol{\delta} = \boldsymbol{p} \odot \boldsymbol{m}$
6:     $g_{\widetilde{m}} = \nabla_{\boldsymbol{\delta}}\mathcal{L} \odot \boldsymbol{p} \odot \sigma'(\widetilde{m})$
7:     $\boldsymbol{p} = \Pi_{\mathcal{S}_p}(\boldsymbol{p} + \alpha \cdot \texttt{sign}(g_{\boldsymbol{p}}))$
8:     $\boldsymbol{d} = g_{\widetilde{m}}/(\|g_{\widetilde{m}}\|_2 + \gamma)$ **if** $\|g_{\widetilde{m}}\|_2 \geq 2 \cdot \gamma$ **else** $0$
9:     $\boldsymbol{m}_{old}, \widetilde{m} = \boldsymbol{m}, \widetilde{m} + \beta \cdot \boldsymbol{d}$
10:    **if** attack succeeds:
11:         break
12:    **if** $\|\Pi_{\mathcal{S}_m}(\sigma(\widetilde{m})) - \boldsymbol{m}_{old}\|_0 \leq 0$ for 3 consecutive iterations:
13:         Random initialize $\widetilde{m}$
14: **end for**
> **Output:** Perturbation: $\boldsymbol{\delta} = \boldsymbol{p} \odot \boldsymbol{m}$

---

### 4.2 SPARSE-AUTOATTACK (SAA)

AutoAttack (AA) (Croce & Hein, 2020) is an ensemble of four diverse attacks for a standardized parameters-free and reliable evaluation of robustness against $l_\infty$ and $l_2$ attacks. Croce & Hein (2021) extends AutoAttack to $l_1$ bounded perturbations. In this work, we further extend this framework to the $l_0$ case. Similar to AA, we include both untargeted and targeted attacks. However, we find the adaptive step size, momentum and difference of logits ratio (DLR) loss function, which are all included in the AutoPGD in AA, do not help to improve the performance in the $l_0$ case. Instead, we run both untargeted sPGD and targeted sPGD (top $C = 9$ incorrect labels) with cross-entropy loss and constant step sizes. In addition, we run sPGD twice for each case: the first time with the sparse gradient $g_{\boldsymbol{p}}$ and the second with the unprojected gradient $\widetilde{g}_{\boldsymbol{p}}$ as described in Section 4.1. we denote the untargeted sPGD as sPGD$_{\text{CE}}$, the targeted one as sPGD$_{\text{CE-T}}$ and their ensemble

as $sPGD_{CE+T}$. To ensure our method has similar computational complexity as AA, we do not include the white-box FAB attack (Croce & Hein, 2019a) in sAA. As for the black-box attack, Square Attack (Andriushchenko et al., 2020) adopted in AA does not have a version for the $l_0$ case. Therefore, we adopt the strong black-box attack Sparse-RS (Croce et al., 2022), which can generate $l_0$ bounded perturbations. We use cascade evaluation to improve the efficiency. Concretely, if we find an adversarial perturbation by one attack to change the model's prediction for one instance, we will consider the model non-robust on this instance and the same instance will not be further evaluated by the other attacks. In this regard, the attacks in sAA are sorted from the ones with low complexity to the ones with high complexity: untargeted sPGD, targeted sPGD and Sparse-RS.

## 4.3 ADVERSARIAL TRAINING

In addition to robustness evaluation, we also explore adversarial training to build robust models against sparse perturbations. In the framework of adversarial training, the attack is used to generate adversarial perturbation in each training iteration, so the attack algorithm should not be too computationally expensive. In this regard, we adopt the untargeted sPGD (Algorithm 1) to generate sparse adversarial perturbations for training. We incorporate sPGD in the framework of vanilla adversarial training (Madry et al., 2017) and TRADES (Zhang et al., 2019b). The corresponding methods are called **sAT** and **sTRADES**, respectively. Since the sparse gradient and the unprojected gradient as described in Section 4.1 induce different exploration-exploitation trade-offs, we randomly select one backward function to generate adversarial perturbations for each mini-batch when using sPGD to generate adversarial perturbations for training the model.

## 5 EXPERIMENTS

In this section, we conduct extensive experiments to compare our attack methods with baselines in evaluating the robustness of various models against $l_0$ bounded perturbations. In addition to effectiveness with an abundant query budget, we also study the efficiency of our methods when we use limited iterations to generate adversarial perturbations. Our proposed methods demonstrate impressive performance in both aspects: sAA, consisting of sPGD and sparse-RS(Croce et al., 2022), has the best attack success rate; sPGD has much better performance than all baselines when using limited iterations, making sPGD feasible for adversarial training and able to obtain the state-of-the-art robust models against $l_0$ bounded perturbations crafted by sAA. In addition, we conduct ablation studies for analysis. The adversarial examples generated by our methods are visually interpretable and presented in Appendix C. Implementation details are deferred to Appendix B.

### 5.1 EVALUATION OF DIFFERENT ATTACK METHODS

First, we compare our proposed sPGD, including $sPGD_{CE}$ and $sPGD_{CE+T}$ as defined in Section 4.2, and sparse-AutoAttack (sAA) with existing white-box and black-box attacks that generate sparse perturbations. Unless specifically mentioned, we evaluate different attack methods based on the models trained on CIFAR-10 (Krizhevsky et al., 2009) and report the robust accuracy with $k = 20$ on the whole test set (see in Table 1). In Appendix A.1, we report more results with $k = 10$, $k = 15$, and the results of models trained on CIFAR-100 (Krizhevsky et al., 2009) in Table 6, 7 and 8, respectively to comprehensively demonstrate the effectiveness of our methods.

**Models:** We select various models to comprehensively evaluate their robustness against $l_0$ bounded perturbations. As a baseline, we train a ResNet-18 (RN-18) (He et al., 2016a) model on clean inputs. For adversarially trained models, we select competitive models that are publicly available, including those trained against $l_\infty$, $l_2$ and $l_1$ bounded perturbations. For the $l_\infty$ case, we include adversarial training with the generated data (GD) (Gowal et al., 2021), the proxy distributions (PORT) (Sehwag et al., 2021), the decoupled KL divergence loss (DKL) (Cui et al., 2023) and strong diffusion models (DM) (Wang et al., 2023). For the $l_2$ case, we include adversarial training with the proxy distributions (PORT) (Sehwag et al., 2021), strong diffusion models (DM) (Wang et al., 2023), helper examples (HAT) (Rade & Moosavi-Dezfooli, 2021) and strong data augmentations (FDA) (Rebuffi et al., 2021). The $l_1$ case is less explored, so we only include $l_1$-APGD adversarial training (Croce & Hein, 2021) and the efficient Fast-EG-$l_1$ (Jiang et al., 2023) for comparison. The network architecture used in these baselines is either ResNet-18 (RN-18), PreActResNet-18 (PRN-18) (He et al.,

2016b) or WideResNet-28-10 (WRN-28) (Zagoruyko & Komodakis, 2016). Finally, we use $PGD_0$ (Croce & Hein, 2019c) in vanilla adversarial training ($PGD_0$-A) and TRADES ($PGD_0$-T), and our proposed sPGD in vanilla adversarial training (sAT) and TRADES (sTRADES) to obtain PRN-18 models to compare with these baselines. Note that the hyperparameters for $PGD_0$-A and $PGD_0$-T are the same as those reported in (Croce & Hein, 2019c).

**Attacks:** We compare our methods with various existing black-box and white-box attacks that generate $l_0$ bounded perturbations. The black-box attacks include CornerSearch (CS) (Croce & Hein, 2019c) and Sparse-RS (RS) (Croce et al., 2022). The white-box attacks include SparseFool (SF) (Modas et al., 2018), $PGD_0$ (Croce & Hein, 2019c) and Sparse Adversarial and Interpretable Attack Framework (SAIF) (Imtiaz et al., 2022). To boost the strength of white-box attacks, we ensemble their untargeted and targeted versions (CE+T) for evaluation, except SF due to its high complexity. For a fair comparison, we keep the number of total iterations of all attacks approximately the same, the details of which are deferred to Appendix B. Note that we report the robust accuracy under CS attack based on only 1000 random test instances due to its prohibitively high computational complexity.

Based on the results in Table 1, we can find that SF attack, $PGD_0$ attack and SAIF attack, including both the untargeted and targeted versions, perform significantly worse than our methods for all the models studied. That is, our proposed sPGD always performs the best among white-box attacks.

Among black-box attacks, CS attack can achieve competitive performance, but it runs dozens of times longer than our method does. Therefore, we focus on comparing our method with RS attack. For $l_1$ and $l_2$ models, our proposed $sPGD_{CE+T}$ significantly outperforms RS attack. By contrast, RS attack outperforms $sPGD_{CE+T}$ for $l_\infty$ models. This indicates gradient masking still exists in this case. Nevertheless, sAA still achieves the best performance, with a considerable margin on top of both RS attack and $sPGD_{CE+T}$. The results indicate the necessity of combining both white-box and black-box attacks for comprehensive robustness evaluation.

In the case of sAT and sTRADES, the models are adversarially trained against sPGD attack. The model also suffers from gradient masking to some extent, which is indicated by the degraded robust accuracy by the strong black-box RS attack. However, Figure 1 (a) illustrates that the performance of RS attack drastically deteriorates with limited iterations (e.g. smaller than 300), so RS is not suitable for adversarial training where the efficiency is required. Despite this, compared with other models in Table 1, the models trained by sAT and sTRADES still show the strongest robustness, indicated by the comprehensive sAA method and all other attack methods. Compared with sAT, sTRADES achieves better performance both in robust accuracy and accuracy on the clean data. Since we have not exhaustively investigated robust learning methods using sPGD, the preliminary results of sAT and sTRADES indicate the potential for further performance improvement. We leave this as future work. Besides models adversarially trained against sPGD, models trained against $PGD_0$ show non-robustness to our attack. Nevertheless, models trained by $l_1$ bounded perturbations are the most robust ones among existing training methods. It could be attributed to the fact that $l_1$ norm is the tightest convex relaxation of $l_0$ norm (Bittar et al., 2021). From a qualitative perspective, $l_1$ attacks also generate relatively sparse perturbations (Jiang et al., 2023), which makes the corresponding model robust to sparse perturbations to some degree.

The results in Table 1 indicate sPGD attack and RS attack can complement each other. Therefore, sAA, an AutoAttack-style attack that ensembles both attacks achieves the state-of-the-art performance on all models. In the implementations, sAA has a similar computational complexity to AutoAttack in $l_\infty$, $l_2$ and $l_1$ cases. Hyper-parameter details are deferred to Appendix B.

## 5.2 COMPARISON UNDER DIFFERENT ITERATION NUMBERS AND SPARSITY LEVELS

In this subsection, we further compare our method sPGD, which is a white-box attack, with RS attack, the strongest black-box attack in Table 1. Specifically, we compare these two attacks under different iteration numbers and sparsity levels.

**Robust Acc. v.s. Iteration Numbers:** As illustrated in Figure 1 (a), although the performances of both $sPGD_{CE}$ and RS attack get improved with more iterations, $sPGD_{CE}$ achieves significantly better performance than RS attack when the iteration number is small, which makes it feasible for adversarial training. Similar to other black-box attacks, the performance of RS attack drastically

Table 1: Robust accuracy of various models on different attacks that generate $l_0$ bounded perturbations, where the sparsity level $k = 20$. The models are trained on CIFAR-10 (Krizhevsky et al., 2009). Note that CornerSearch (CS) is evaluated on 1000 samples due to its high computational complexity.

| Model | Network | Clean | CS | RS | SF | PGD$_0$ (CE+T) | SAIF (CE+T) | sPGD$_{CE}$ | sPGD$_{CE+T}$ | sAA |
|---|---|---|---|---|---|---|---|---|---|---|
| Vanilla | RN-18 | 93.9 | 1.2 | 0.0 | 17.5 | 0.4 | 3.2 | 0.0 | 0.0 | **0.0** |
| $l_\infty$-adv. trained, $\epsilon = 8/255$ | | | | | | | | | | |
| GD | PRN-18 | 87.4 | 26.7 | 11.0 | 52.6 | 25.2 | 40.4 | 19.7 | 15.7 | **9.2** |
| PORT | RN-18 | 84.6 | 27.8 | 14.6 | 54.5 | 21.4 | 42.7 | 20.9 | 16.1 | **10.8** |
| DKL | WRN-28 | 92.2 | 33.1 | 11.0 | 54.0 | 29.3 | 41.1 | 22.5 | 16.6 | **9.6** |
| DM | WRN-28 | 92.4 | 32.6 | 10.3 | 49.4 | 26.9 | 38.5 | 21.5 | 15.9 | **9.0** |
| $l_2$-adv. trained, $\epsilon = 0.5$ | | | | | | | | | | |
| HAT | PRN-18 | 90.6 | 34.5 | 20.5 | 56.3 | 22.5 | 49.5 | 13.2 | 10.0 | **9.3** |
| PORT | RN-18 | 89.8 | 30.4 | 18.7 | 55.0 | 17.2 | 48.0 | 10.5 | 7.5 | **7.1** |
| DM | WRN-28 | 95.2 | 43.3 | 23.4 | 59.2 | 31.8 | 59.6 | 20.9 | 15.5 | **14.1** |
| FDA | WRN-28 | 91.8 | 43.8 | 25.8 | 64.2 | 25.5 | 57.3 | 26.4 | 21.3 | **18.4** |
| $l_1$-adv. trained, $\epsilon = 12$ | | | | | | | | | | |
| $l_1$-APGD | PRN-18 | 80.7 | 32.3 | 33.1 | 65.4 | 39.8 | 55.6 | 24.0 | 20.0 | **19.5** |
| Fast-EG-$l_1$ | PRN-18 | 76.2 | 35.0 | 31.5 | 60.8 | 37.1 | 50.0 | 24.4 | 20.0 | **19.4** |
| $l_0$-adv. trained, $k = 20$ | | | | | | | | | | |
| PGD$_0$-A | PRN-18 | 76.2 | 1.3 | 0.1 | 17.1 | 0.0 | 1.3 | 0.0 | 0.0 | **0.0** |
| PGD$_0$-T | PRN-18 | 78.2 | 0.7 | 0.1 | 16.6 | 0.0 | 0.7 | 0.0 | 0.0 | **0.0** |
| **sAT** | PRN-18 | 85.8 | 48.1 | 45.1 | 85.2 | 79.7 | 77.1 | 78.2 | 76.8 | **44.6** |
| **sTRADES** | PRN-18 | 87.2 | 55.0 | 52.1 | 86.3 | 82.2 | 79.2 | 79.5 | 77.9 | **52.0** |

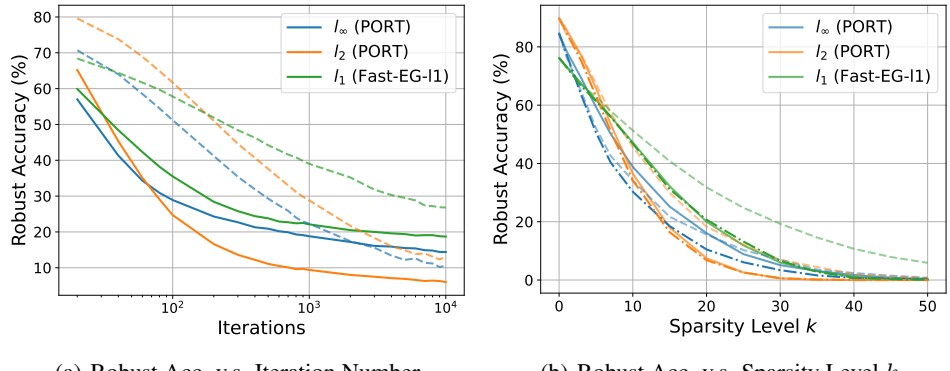

(a) Robust Acc. v.s. Iteration Number      (b) Robust Acc. v.s. Sparsity Level $k$

Figure 1: Comparison between sPGD and RS attack under different iterations and sparsity levels. PORT ($l_\infty$ and $l_2$) (Sehwag et al., 2021) and Fast-EG-$l_1$ (Jiang et al., 2023) are evaluated. **The results of sPGD and RS attack are shown in solid lines and dotted lines, respectively.** (a) Curves of robust accuracy v.s. iterations. sPGD$_{CE}$ is evaluated here. The sparsity level $k$ is set to 20. The total iteration number ranges from 20 and 10000. For better visualization, the x-axis is shown in the log scale. (b) Curves of robust accuracy v.s. sparsity level $k$. sPGD$_{CE+T}$ is evaluated here. The number of total iterations is set to 3000. The results of sAA are shown in dash-dot line.

deteriorates when the query budget is limited. Specifically, when the iteration number is smaller than 2000, which is still considerably large, sPGD$_{CE}$ achieves better performance than RS attack on all models studied in Figure 1 (a). In addition, their performance gaps also increase as the iteration number decreases. Besides the untargeted version, the comparison between sPGD$_{CE+T}$ and RS is presented in Figure 2 of Appendix A.2, where the observations are consistent with Figure 1 (a).

**Robust Acc. v.s. Sparsity Level $k$:** We can observe from Figure 1 (b) that RS attack shows slightly better performance only on the $l_\infty$ model and when $k$ is small. The search space for the perturbed features is relatively small when $k$ is small, which facilitates heuristic black-box search methods like

RS attack. As $k$ increases, $sPGD_{CE+T}$ outperforms RS attack in all cases until both attacks achieve almost $100\%$ attack success rate.

## 5.3 ABLATION STUDIES

We conduct ablation studies to validate the effectiveness of our methods. We focus on CIFAR10 and the sparsity level $k = 20$. Unless specified, we use the same configurations as in Table 1.

**Components of $sPGD_{CE+T}$:** We first validate the effectiveness of each component in $sPGD_{CE+T}$. The result is reported in Table 2. We observe that naively decomposing the perturbation $\delta$ by $\delta = p \odot m$ and updating them separately can deteriorate the performance. By contrast, the performance significantly improves when we update the mask $m$ by its continuous alternative $\widetilde{m}$ and $l_0$ ball projection. This indicates that introducing $\widetilde{m}$ greatly mitigates the challenges in optimizing discrete variables. Moreover, the results in Table 2 indicate the performance can be further improved by the random reinitialization mechanism, which encourages exploration and avoids trapping in a local optimum. In Appendix A.3, we compare the performance when we use different step sizes for the magnitude tensor $p$ and the sparsity mask $m$. The results in Table 10 and 11 of Appendix A.3 indicate that the performance of our proposed method is quite consistent under different choices of step sizes, which facilities hyper-parameter selection for practitioners.

Table 2: Ablation study of each component in $sPGD_{CE+T}$ in terms of robust accuracy. The model is trained by Fast-EG-$l_1$ (Jiang et al., 2023).

| Ablations | Robust Acc. |
|---|---|
| Baseline (PGD$_0$ w/o restart) | 49.5 |
| + Decomposition: $\delta = p \odot m$ | 50.0 (+0.5) |
| + Continuous mask $\widetilde{m}$ | 22.2 (-27.3) |
| + Random reinitialization | 20.0 (-29.5) |

Table 3: Comparison between different backward functions in $sPGD_{CE+T}$ on various models.

| Model | Projected | Unprojected |
|---|---|---|
| Vanilla | **0.1** | 0.2 |
| $l_\infty$ (PORT) | **16.1** | 16.5 |
| $l_2$ (PORT) | 14.0 | **7.4** |
| $l_1$ (Fast-EG-$l_1$) | 28.4 | **20.0** |
| $l_0$ (sTRADES) | **77.9** | 81.5 |

**Different Backward Functions:** We also validate the effectiveness of different backward functions, i.e., the sparse gradient and the unprojected gradient, on various models. The results in Table 3 indicate the two backward functions are complementary: neither of them is better in all cases. As a result, our proposed sPGD and sAA use both the sparse and the unprojected gradients to ensure a more comprehensive evaluation of robustness. In adversarial training, we randomly select one backward function to construct adversarial examples for training to ensure the model's robustness against attacks based on different backward functions.

**Adversarial training:** We conduct preliminary exploration on adversarial training against sparse perturbations. Table 4 demonstrates the robust accuracy of models trained by sPGD with different iteration numbers. The results indicate that the robust accuracy increases with the number of iterations first and then saturates. To balance the performance and efficiency, we use 100-iteration sPGD by default. Table 5 demonstrates the performance when we use different backward functions. The policies include always using the sparse gradient (Sparse), always using the unprojected gradient (Unproj.), alternatively using both backward functions every 5 epochs (Alter.) and randomly selecting backward functions (Rand.). The results indicate that randomly selecting backward functions has the best performance, both for sAT and sTRADES. Comprehensively comparing different adversarial training variants and designing the loss objective function in the context of sPGD to further boost the model robustness against sparse perturbations are left as future work.

Table 4: Ablation study on iteration numbers of attacks during adversarial training. The robust accuracy is obtained through sAA.

| Method | 10 | 50 | 100 | 150 | 200 |
|---|---|---|---|---|---|
| sAT | 43.1 | 40.7 | 44.8 | **45.7** | 44.4 |
| sTRADES | 38.7 | 48.5 | 52.0 | **53.2** | 53.1 |

Table 5: Different backward functions and losses during adversarial training. The robust accuracy is obtained through sAA.

| Method | Sparse | Unproj. | Alter. | Rand. |
|---|---|---|---|---|
| sAT | 41.5 | 27.2 | 38.9 | **44.8** |
| sTRADES | 47.6 | 42.5 | 49.8 | **52.0** |

# 6 CONCLUSION

In this paper, we propose an effective and efficient white-box attack to generate sparse perturbations bounded by the $l_0$ norm. Based on that, we further propose an ensemble of both white-box and black-box attacks for reliable $l_0$ robustness evaluation. Our proposed white-box attack, due to its efficiency, can also be used in adversarial training to obtain robust models against sparse perturbations. Our attack methods outperform the state-of-the-art sparse attacks. The robust models obtained by our training method demonstrate the best robust accuracy. Our future work will focus on constructing more efficient adversarial training algorithms against sparse perturbations.

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

# A  ADDITIONAL EXPERIMENTS

## A.1  RESULTS OF DIFFERENT SPARSITY LEVELS AND DIFFERENT DATASETS

In this subsection, we present the robust accuracy on CIFAR-10 with the sparsity level $k = 10$ and $k = 15$, those on CIFAR-100 with $k = 10$, those on ImageNet-100 (Deng et al., 2009) with $k = 200$ in Table 6, 7, 8 and 9, respectively. Since CIFAR-100 and ImageNet-100 have 100 categories, we evaluate only the top 9 incorrect classes with the highest confidence scores in targeted attacks. Note that $PGD_0$-A and $PGD_0$-T are evaluated in this section, due to their non-robustness to our attack. The observations with different sparsity levels and on different datasets are consistent with those in Table 1, which indicates the effectiveness of our method.

In addition, we evaluate our methods on MNIST (Deng, 2012) with the sparsity level $k = 30$. We train a LeNet model (LeCun et al., 1998) under various settings, including the clean inputs, $l_\infty$ bounded perturbation with the magnitude $\epsilon = 0.3$ and $l_0$ bounded perturbations with the sparsity level $k = 30$. The model trained by sTRADES has a competitive robust accuracy of $45.8\%$ under sAA. By contrast, the vanilla model and the $l_\infty$ models only have trivial performance (i.e., $0\%$).

Table 6: Robust accuracy of various models on different attacks that generate $l_0$ bounded perturbations, where the sparsity level $k = 10$. The models are trained on CIFAR-10 (Krizhevsky et al., 2009). Note that CornerSearch (CS) is evaluated on 1000 samples due to its high computational complexity.

| Model | Network | Clean | CS | RS | SF | $PGD_0$ (CE+T) | SAIF (CE+T) | $sPGD_{CE}$ | $sPGD_{CE+T}$ | sAA |
|---|---|---|---|---|---|---|---|---|---|---|
| Vanilla | RN-18 | 93.9 | 3.2 | 1.5 | 40.6 | 11.5 | 31.8 | 4.1 | 2.5 | **1.1** |
| $l_\infty$-adv. trained, $\epsilon = 8/255$ | | | | | | | | | | |
| GD | PRN-18 | 87.4 | 36.8 | 31.0 | 69.9 | 50.3 | 63.0 | 44.3 | 38.9 | **28.6** |
| PORT | RN-18 | 84.6 | 36.7 | 33.5 | 70.7 | 46.1 | 62.6 | 45.4 | 38.8 | **30.3** |
| DKL | WRN-28 | 92.2 | 40.9 | 30.4 | 71.9 | 54.2 | 64.6 | 47.8 | 41.3 | **28.8** |
| DM | WRN-28 | 92.4 | 38.7 | 28.6 | 68.7 | 52.7 | 62.5 | 46.6 | 40.1 | **27.5** |
| $l_2$-adv. trained, $\epsilon = 0.5$ | | | | | | | | | | |
| HAT | PRN-18 | 90.6 | 47.3 | 47.2 | 74.6 | 53.5 | 71.4 | 44.6 | 40.1 | **38.5** |
| PORT | RN-18 | 89.8 | 46.8 | 45.1 | 74.2 | 50.4 | 70.9 | 41.2 | 36.5 | **34.9** |
| DM | WRN-28 | 95.2 | 57.8 | 54.3 | 78.3 | 65.5 | 80.9 | 58.0 | 51.9 | **48.3** |
| FDA | WRN-28 | 91.8 | 55.0 | 54.3 | 79.6 | 58.6 | 77.5 | 58.2 | 52.6 | **48.7** |
| $l_1$-adv. trained, $\epsilon = 12$ | | | | | | | | | | |
| $l_1$-APGD | PRN-18 | 80.7 | 51.4 | 54.9 | 74.3 | 60.7 | 68.1 | 52.3 | 49.1 | **48.3** |
| Fast-EG-$l_1$ | PRN-18 | 76.2 | 49.7 | 50.6 | 69.7 | 56.7 | 63.2 | 50.8 | 47.2 | **46.1** |
| $l_0$-adv. trained, $k = 20$ | | | | | | | | | | |
| **sAT** | PRN-18 | 85.8 | **62.5** | 65.7 | 85.3 | 82.4 | 81.5 | 81.7 | 80.7 | 65.7 |
| **sTRADES** | PRN-18 | 87.2 | **67.5** | 70.1 | 86.5 | 84.5 | 83.6 | 83.0 | 82.2 | 70.1 |

## A.2  COMPARISON BETWEEN $sPGD_{CE+T}$ AND SPARSE-RS UNDER DIFFERENT ITERATIONS

In Figure 1 (a), we compare $sPGD_{CE}$ and sparse-RS (RS) under different iterations, which demonstrates $sPGD_{CE}$ achieves significantly better performance than RS attack when the iteration number is small, which makes it feasible for adversarial training. Here, we additionally compare $sPGD_{CE+T}$ and RS under different iterations in Figure 2. The observed phenomenon is consistent with that in Figure 1 (a).

## A.3  STEP SIZE IN SPGD

As shown in Table 10 and 11, the robust accuracy does not vary significantly with different step sizes. It indicates the robustness of our method to different hyper-parameter choices. In practice, We set $\alpha$ and $\beta$ to 0.25 and $0.25 \times \sqrt{h \times w}$, respectively. Note that $h$ and $w$ denote the height and width of the image, respectively, which are both 32 in CIFAR-10.

Table 7: Robust accuracy of various models on different attacks that generate $l_0$ bounded perturbations, where the sparsity level $k = 15$. The models are trained on CIFAR-10 (Krizhevsky et al., 2009). Note that CornerSearch (CS) is evaluated on 1000 samples due to its high computational complexity.

| Model | Network | Clean | CS | RS | SF | PGD$_0$ (CE+T) | SAIF (CE+T) | sPGD$_{CE}$ | sPGD$_{CE+T}$ | sAA |
|---|---|---|---|---|---|---|---|---|---|---|
| Vanilla | RN-18 | 93.9 | 1.6 | 0.1 | 25.3 | 2.1 | 12.0 | 0.3 | 0.1 | **0.0** |
| $l_\infty$-adv. trained, $\epsilon = 8/255$ | | | | | | | | | | |
| GD | PRN-18 | 87.4 | 30.5 | 18.7 | 61.1 | 36.0 | 51.3 | 30.1 | 24.9 | **16.9** |
| PORT | RN-18 | 84.6 | 30.8 | 22.1 | 62.1 | 31.4 | 52.1 | 31.2 | 25.2 | **18.1** |
| DKL | WRN-28 | 92.2 | 35.3 | 18.3 | 62.5 | 41.2 | 52.3 | 33.2 | 26.7 | **16.9** |
| DM | WRN-28 | 92.4 | 34.8 | 17.0 | 57.9 | 38.5 | 49.4 | 31.9 | 25.5 | **15.8** |
| $l_2$-adv. trained, $\epsilon = 0.5$ | | | | | | | | | | |
| HAT | PRN-18 | 90.6 | 38.9 | 31.4 | 65.3 | 35.4 | 60.2 | 25.4 | 21.0 | **20.0** |
| PORT | RN-18 | 89.8 | 36.8 | 29.4 | 64.3 | 30.6 | 59.7 | 21.9 | 18.0 | **17.1** |
| DM | WRN-28 | 95.2 | 48.5 | 35.6 | 68.2 | 47.5 | 70.9 | 36.9 | 26.7 | **27.1** |
| FDA | WRN-28 | 91.8 | 47.8 | 38.1 | 71.8 | 40.1 | 68.2 | 40.4 | 34.7 | **30.8** |
| $l_1$-adv. trained, $\epsilon = 12$ | | | | | | | | | | |
| $l_1$-APGD | PRN-18 | 80.7 | 41.3 | 43.5 | 70.3 | 50.5 | 62.3 | 36.9 | 33.0 | **32.1** |
| Fast-EG-$l_1$ | PRN-18 | 76.2 | 40.7 | 40.3 | 64.9 | 46.7 | 56.9 | 36.5 | 32.2 | **31.2** |
| $l_0$-adv. trained, $k = 20$ | | | | | | | | | | |
| **sAT** | PRN-18 | 85.8 | **54.4** | 55.0 | 85.2 | 80.9 | 79.5 | 80.2 | 79.0 | 54.9 |
| **sTRADES** | PRN-18 | 87.2 | 61.2 | 60.9 | 86.4 | 83.2 | 81.3 | 81.3 | 80.2 | **60.9** |

Table 8: Robust accuracy of various models on different attacks that generate $l_0$ bounded perturbations, where the sparsity level $k = 10$. The models are trained on CIFAR-100 (Krizhevsky et al., 2009). To accelerate the evaluation, targeted attacks on the top-9 classes with the highest confidence scores are adopted. Note that CornerSearch (CS) is evaluated on 1000 samples due to its high computational complexity.

| Model | Network | Clean | CS | RS | SF | PGD$_0$ (CE+T) | SAIF (CE+T) | sPGD$_{CE}$ | sPGD$_{CE+T}$ | sAA |
|---|---|---|---|---|---|---|---|---|---|---|
| Vanilla | RN-18 | 74.3 | 1.6 | 0.3 | 20.1 | 1.9 | 9.0 | 0.8 | 0.4 | **0.2** |
| $l_\infty$-adv. trained, $\epsilon = 8/255$ | | | | | | | | | | |
| HAT | PRN-18 | 61.5 | 12.6 | 11.3 | 39.1 | 19.1 | 26.8 | 18.9 | 15.4 | **10.4** |
| FDA | PRN-18 | 56.9 | 16.3 | 13.5 | 42.2 | 23.0 | 30.7 | 22.5 | 19.0 | **12.9** |
| DKL | WRN-28 | 73.8 | 12.4 | 7.7 | 44.9 | 20.9 | 26.5 | 17.2 | 13.9 | **7.3** |
| DM | WRN-28 | 72.6 | 14.0 | 9.6 | 46.2 | 23.4 | 29.8 | 20.9 | 16.7 | 9.0 |
| $l_1$-adv. trained, $\epsilon = 6$ | | | | | | | | | | |
| $l_1$-APGD | PRN-18 | 63.2 | 22.7 | 26.1 | 47.7 | 33.0 | 43.5 | 25.6 | 21.6 | **20.7** |
| Fast-EG-$l_1$ | PRN-18 | 59.4 | 21.5 | 21.0 | 44.8 | 30.6 | 39.5 | 24.3 | 20.6 | **19.4** |
| $l_0$-adv. trained, $k = 10$ | | | | | | | | | | |
| **sAT** | PRN-18 | 56.2 | **33.6** | 36.0 | 55.3 | 51.7 | 51.9 | 51.0 | 49.8 | 35.7 |
| **sTRADES** | PRN-18 | 60.5 | **38.7** | 39.0 | 59.8 | 54.9 | 52.4 | 55.4 | 54.2 | 38.8 |

Table 10: Robust accuracy at different step sizes $\alpha$ for magnitude $p$. The evaluated attack is sPGD$_{CE+T}$. The model is Fast-EG-$l_1$ (Jiang et al., 2023) trained on CIFAR-10 (Krizhevsky et al., 2009).

| $\alpha$ | $\frac{1}{16}$ | $\frac{1}{8}$ | $\frac{1}{4}$ | $\frac{1}{2}$ | $\frac{3}{4}$ | 1 |
|---|---|---|---|---|---|---|
| Acc. | 20.1 | 20.1 | **20.0** | 20.2 | 20.2 | 20.2 |

Table 11: Robust accuracy at different step sizes $\beta$ for mask $m$. The evaluated attack is sPGD$_{CE+T}$. The model is Fast-EG-$l_1$ (Jiang et al., 2023) trained on CIFAR-10 (Krizhevsky et al., 2009).

| $\beta$ | 2 | 4 | 8 | 16 | 24 | 32 |
|---|---|---|---|---|---|---|
| Acc. | 20.1 | 20.1 | **20.0** | 20.1 | 20.2 | 20.2 |

# B  IMPLEMENTATION DETAILS

Table 9: Robust accuracy of various models on different attacks that generate $l_0$ bounded perturbations, where the sparsity level $k = 200$. The models are trained on ImageNet-100 (Deng et al., 2009), and ResNet34 (RN-34) (He et al., 2016a) is used as the network architecture. Our sAT model is trained with the sparsity level $k = 200$ and the iteration number of the attack $t = 20$. Note that all methods are evaluated on 500 samples, and CornerSearch (CS) is not evaluated here due to the high computational complexity.

| Model | Network | Clean | CS | RS | SF | $PGD_0$ (CE+T) | SAIF (CE+T) | sPGD | $sPGD_{CE+T}$ | sAA |
|---|---|---|---|---|---|---|---|---|---|---|
| Vanilla | RN-34 | 83.0 | - | 0.6 | 5.8 | 9.8 | 0.6 | 0.2 | 0.0 | **0.0** |
| $l_1$-adv. trained, $\epsilon = 72$ | | | | | | | | | | |
| Fast-EG-$l_1$ | RN-34 | 69.2 | - | 48.0 | 43.4 | 50.2 | 43.0 | 23.8 | 19.4 | **19.4** |
| $l_0$-adv. trained, $k = 200$ | | | | | | | | | | |
| **sAT** | RN-34 | 84.8 | - | 57.8 | 83.6 | 80.8 | 62.4 | 76.4 | 73.4 | **57.8** |

In experiments, we mainly focus on the cases of the sparsity of perturbations $k = 10, 15$ and 20, where $k = ||\sum_{i=1}^{c} \boldsymbol{\delta}^{(i)}||_0$ or $||\boldsymbol{m}||_0$, $\boldsymbol{\delta}^{(i)} \in \mathbb{R}^{h \times w}$ is the $i$-th channel of perturbation $\boldsymbol{\delta} \in \mathbb{R}^{h \times w \times c}$, and $\boldsymbol{m} \in \mathbb{R}^{h \times w \times 1}$ is the sparsity mask in the decomposition of $\boldsymbol{\delta} = \boldsymbol{p} \odot \boldsymbol{m}, \boldsymbol{p} \in \mathbb{R}^{h \times w \times c}$ denotes the magnitude of perturbations.

**SparseFool** (Modas et al., 2018): We apply SparseFool following the official implementation and use the default value of the sparsity parameter $\lambda = 3$. The maximum iterations per sample is set to 3000 to ensure fair comparison among attacks in Table 1. Finally, the perturbation generated by SparseFool is projected to the $l_0$ ball to satisfy the adversarial budget.

**CornerSearch** (Croce & Hein, 2019c): For CornerSearch, we set the hyperparameters as following: $N = 100, N_{iter} = 3000$, where $N$ is the sample size of the one-pixel perturbations, $N_{iter}$ is the number of queries. For bot CIFAR-10 and CIFAR-100 datasets, we evaluate the robust accuracy on 1000 test instances due to its prohibitively high computational complexity.

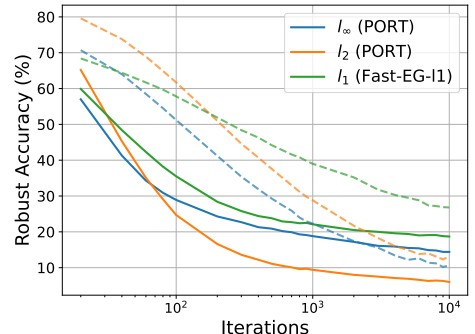

Figure 2: Curves of robust accuracy v.s. iterations. $sPGD_{CE+T}$ is evaluated here. The sparsity level $k$ is set to 20. The total iteration number ranges from 20 and 10000. PORT ($l_\infty$ and $l_2$) (Sehwag et al., 2021) and Fast-EG-$l_1$ (Jiang et al., 2023) are evaluated. The results of sPGD and RS attack are shown in solid lines and dotted lines, respectively. For better visualization, the x-axis is shown in log scale.

**Sparse-RS** (Croce et al., 2022): For Sparse-RS, we set $\alpha_{init} = 0.3$, which controls the set of pixels changed in each iteration. We only report the results of untargeted attacks with the maximum queries limited to 3000.

**$PGD_0$** (Croce & Hein, 2019c): For $PGD_0$, we include both untargeted attack and targeted attacks on the top-9 incorrect classes with the highest confidence scores. We set the step size to $\eta = 120000/255$. The iteration numbers of each attack are 300. Besides, 5 restarts are adopted to further boost the performance.

**SAIF** (Imtiaz et al., 2022): Similar to $PGD_0$, we apply both untargeted attack and targeted attacks on the top-9 incorrect classes with 300 iterations per attack. We adopt the same $l_\infty$ norm constraint for the magnitude tensor $p$ as in sPGD.

**sparse-PGD (sPGD)**: Cross-entropy loss is adopted as the loss function of both untargeted and targeted versions of our method. The step size for magnitude $\boldsymbol{p}$ is set $\alpha = 1/4$; the step size for continuous mask $\widetilde{\boldsymbol{m}}$ is set $\beta = 1/4 \times \sqrt{h \times w}$, where $h$ and $w$ are the height and width of the input image $\boldsymbol{x} \in \mathbb{R}^{h \times w \times c}$, respectively. The small constant $\gamma$ to avoid numerical error is set to $1 \times 10^{-10}$. The number of iterations is 300 for all datasets to ensure fair comparison among attacks in Table 1.

**sparse-AutoAttack (sAA)**: It is a cascade ensemble of five different attacks, i.e., **a)** untargeted sPGD with sparse gradient, **b)** untargeted sPGD with unprojected gradient, **c)** targeted sPGD with

sparse gradient,**d)** targeted sPGD with unprojected gradient and **e)** untargeted Sparse-RS. The hyper-parameters of sPGD are the same as those listed in the last paragraph. We apply targeted attacks on the top-9 incorrect classes with the highest confidence scores. The iteration number of untargeted Sparse-RS is 3000 for all datasets.

**Adversarial Training**: sPGD and $PGD_0$ are adopted as the attack during the training phase, the number of iterations is 100, and the backward function is randomly selected from the two different backward functions for each batch. The sparsity of perturbations are 20 and 10 for CIFAR-10 and CIFAR-100, respectively. We use PreactResNet18 (He et al., 2016b) with softplus activation (Dugas et al., 2000) for experiments in CIFAR-10 and CIFAR-100. We train the model for 40 epochs, the training batch size is 128. The optimizer is SGD with a momentum factor of 0.9 and weight decay factor of $5 \times 10^{-4}$. The learning rate is initialized to 0.05 and is divided by a factor of 10 at the 30th epoch and the 35th epoch.

## C  VISUALIZATION OF SOME ADVERSARIAL EXAMPLES

We show some adversarial examples with different sparsity levels of perturbation in Figure 3, 4, 5. The attack is sPGD, and the model is Fast-EG-$l_1$ (Jiang et al., 2023) trained on CIFAR-10. We can observe that most of the perturbed pixels are located in the foreground of images. It is consistent with the intuition that the foreground of an image contains most of the semantic information.

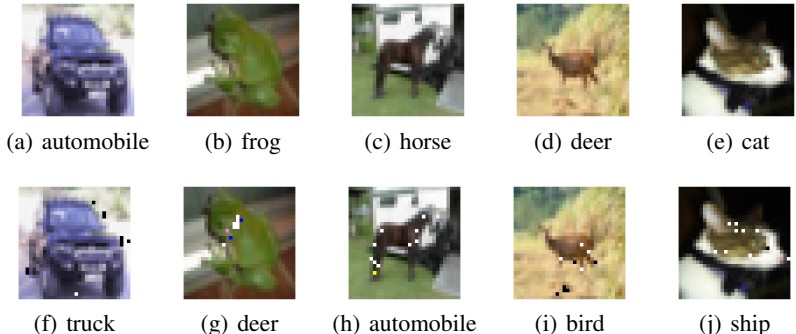

(a) automobile    (b) frog    (c) horse    (d) deer    (e) cat

(f) truck    (g) deer    (h) automobile    (i) bird    (j) ship

Figure 3: Clean images (first row) from the test set of CFIFAR-10 and their corresponding adversarial samples (second row) by sPGD. The attack is sPGD with sparsity level $k = 10$. The model is Fast-EG-$l_1$ trained on CIFAR-10. The predictions given by the model are listed below the images.

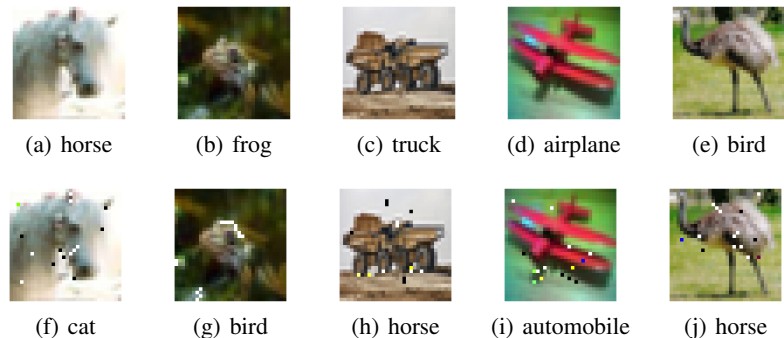

(a) horse    (b) frog    (c) truck    (d) airplane    (e) bird

(f) cat    (g) bird    (h) horse    (i) automobile    (j) horse

Figure 4: Clean images (first row) from the test set of CFIFAR-10 and their corresponding adversarial samples (second row) by sPGD. The attack is sPGD with sparsity level $k = 15$. The model is Fast-EG-$l_1$ trained on CIFAR-10. The predictions given by the model are listed below the images.

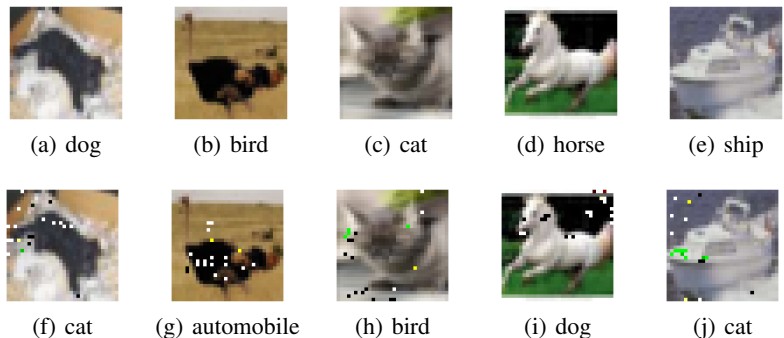

Figure 5: Clean images (first row) from the test set of CFIFAR-10 and their corresponding adversarial samples (second row) by sPGD. The attack is sPGD with sparsity level $k = 20$. The model is Fast-EG-$l_1$ trained on CIFAR-10. The predictions given by the model are listed below the images.

