# OpenReview forum: "Sparse-PGD: An Effective and Efficient Attack for $l_0$ Bounded Adversarial Perturbation"
_ICLR.cc/2024/Conference — Submitted to ICLR 2024_

### Official Review · Reviewer_RqW9 · 2023-10-22

**Soundness:** 2 fair
**Presentation:** 2 fair
**Contribution:** 1 poor
**Rating:** 3
**Confidence:** 5

**Summary:**

This paper proposes a method for creating sparse adversarial perturbations. The authors evaluate the approach comparing with existing sparse image-specific attacks, against model robust to $\ell_\infty$, $\ell_2$, and $\ell_1$ perturbations.

**Strengths:**

+ the attacks seem to be run correctly in the evaluation
+ tested against robust models
+ interesting approach to achieve sparse perturbations

**Weaknesses:**

- experimental evaluation should be improved
- contributions are not fully supported by the experimental evidence and should be clarified

**Questions:**

Overall, the paper is easy to read and well written. The proposed contribution is significant, however the claims should be supported better by the experimental evidence.

**Experimental evaluation should be improved.** The authors claim the approach is explained with image classification as an example, but the approach should be applicable to any kind of data. This is inconsistent with how the method is evaluated. In fact, the authors write in the introduction:

> For image inputs, we consider the pixel sparsity, which is more meaningful than feature sparsity and consistent with existing works (Croce & Hein, 2019c; Croce et al., 2022). That is, a pixel is considered perturbed if any of its channel is perturbed, and sparse perturbation means few pixels are perturbed.

So this means that a value of perturbation equal to x corresponds to x pixels changed, but each pixel might contain up to three features. This is written only in the introduction, which makes the evaluation metrics used later for the experiments unclear.
Moreover, it would be interesting to see the results of this method without this additional constraint. The approach can be still developed, simply by creating a mask for every channel. However, removing this limit would make the attack comparable with many other white-box sparse attacks, including:

* EAD https://arxiv.org/abs/1709.04114
* VFGA https://arxiv.org/abs/2011.12423
* PDPGD https://arxiv.org/abs/2106.01538
* BB https://arxiv.org/abs/1907.01003
* FMN https://arxiv.org/abs/2102.12827

**Unclear difference with SAIF.** The authors state that the attack method is similar to the SAIF attack (beginning of sect. 4.1). However, they don't explain clearly what the difference is and what they add to this similar attack to make it perform better. This should be discussed in sect. 4.1

---

> ### Author Response · Authors · 2023-11-22
>
> Dear reviewer RqW9,
>
> We thank reviewer RqW9 for the helpful review! We are glad that you agree that our proposed contribution is significant. In response to your questions, we offer the following point-to-point answers:
>
> 1. Experimental evaluation should be improved. The authors claim the approach is explained with image classification as an example, but the approach should be applicable to any kind of data. This is inconsistent with how the method is evaluated.
>
>     **Reply**: To help illustrate the formulation, we used image classification as an example and evaluated the proposed attack on CIFAR-10, CIFAR-100 and ImagNet-100. Similar to previous attacks [1 - 3], we don't have any constraint on the input. As a result, the method can be applicable to any data.  Besides, all these datasets are intensively used and benchmarked in the existing literature [1 - 4] to evaluate the model's robustness.  Lastly, since it is a novel attack method, rather than empirically proving it is applicable to some specific application in some specific data domain (although we agree that it will be a good potential future work), **our main contribution** is general: that is, our attack method empirically outperforms the state-of-the-art sparse attack (including both white-box and black-box attacks) and those models adversarially trained by the attack show the strongest robustness against various sparse attacks.
>
> 2. It would be interesting to see the results of this method without this additional constraint. The approach can be still developed, simply by creating a mask for every channel.
>
>     **Reply**: Thanks for pointing this out, we will consider your comments and try to improve our work in future work.
>
> 3. Unclear difference with SAIF
>
>     **Reply**: For the attack process, though we adopt a similar decomposition method to the perturbation to SAIF=, **our attack design is significantly different from the previous work in two perspectives:**
>     1. Instead of using an $\ell_1$ convex surrogate for the $l_0$ norm constraint of the sparsity mask as done in SAIF, we introduce a continuous mask $\widetilde{m}$ and project it back to the feasible set. To prevent the magnitude of $\widetilde{m}$ from being too big, we add a sigmoid function before projecting $\widetilde{m}$ to the binary mask $m$. This **new but simple** strategy has proven to be effective as shown in the ablation studies (Table 2, Sec 5.3) where we observed a huge robust accuracy drop from 49.5\% to 22.2\%. Moreover, we further improve the exploration capability for the mask $m$ by a random reinitialization mechanism, which is shown to improve the performance in the ablation study. This design alleviates the problem of local convergence, which is reported in SAIF.
>     2. The ablation studies (Table 2, Sec 5.3) also indicated naively decomposing the perturbation by $\delta = p \odot m$ can **deteriorate the performance**, which further corroborates the effectiveness of our attack designs.

---

### Official Review · Reviewer_DCtb · 2023-10-31

**Soundness:** 1 poor
**Presentation:** 2 fair
**Contribution:** 3 good
**Rating:** 3
**Confidence:** 4

**Summary:**

In this paper, the authors investigate the performance of Sparse-PGD, an l0 attack for crafting adversarial examples. Specifically, the authors note how little attention there has been on evaluating the robustness of machine learning models based on l0 threat models. To this end, the authors propose an attack that is specifically optimized for this threat model, borrowing ideas from SAIF. Their method, Sparse-PGD, is built from a magnitude tensor and a sparsity mask, whose design attempt to tackle known problems in l0-based optimization with convergence and gradient explosion. In their evaluation, they compare their attacks against a variety of other attacks and (adversarially trained) models and demonstrate compelling results. The paper concludes with an ablation study on varies components of their attack and a brief experiment on adversarial training.

**Strengths:**

**Significance/Originality**- $\ell_0$-based attacks have received much less attention than other $\ell_p$ threat models, but represents more realistic threat models in many domains such as network data

**Quality**- Explorations on adversarial training and a broad set of baselines gives a good measure of attack performance.

**Clarity**- Background is well-written, gives a good summary of the field of AML and the various threat models, making it appealing to a broader audience

**Weaknesses:**

* Optimization is unclear - Section 4.1 requires additional details. Arguments are made concerning when a relaxation is necessary (i.e., through a projection), yet later it is claimed that the relaxation exhibits deficiencies, so the original optimization is used instead. After reading 4.1, it is unclear what optimization sPGD actually entails and what is used in the evaluation.
* Evaluation methodology - There are many important details are not present in the evaluation and necessary plots are missing (see questions for details)
* Contribution of attacks introduced in this work is unclear - It does not seem appropriate to add Sparse-RS as part of sAA, given that Sparse-RS is used verbatim from prior work. The evaluation should only include the contributions made in this work.
* Incomplete characterization of l0-based attacks - JSMA (Papernot, 2016) is not mentioned or evaluated against, even though it is the first l0-based attack

**Questions:**

Thank you for your contribution to ICLR. It was an interesting read. Below, I summarize some of my main questions concerning this work.

1. Section 4.1 can be confusing at times - Section 4.1 should be revisited, given that there seem to be inconsistencies in the motivation of certain decisions and the optimization itself is unclear. Specifically: (a) for updating the magnitude tensor, are p and delta the same variable? (b) Why is the l2-norm of the loss taken in (5)? (c) For updating the sparsity mask, what is gradient ascent performed on? (d) it is unclear what, "Since elements in m are 0 or 1, we use sigmoid to normalize elements in m-tilde to be 0 or 1" is trying to say; aren't elements in m in [0, 1] because of (6)? (e) the argument that projection on the binary set Sm is discarded because coordinate descent is suboptimal is unclear; why is such a projection introduced to be later argued as suboptimal and thus discarded? (In fact, this observation is stated twice) (f) it is unclear where the projection onto the binary set Sm is used in gp and why it is used in tandem with gp-tilde if gp exhibits both non-convergence and gradient explosions, and (g) there are many terms that are co-dependent with other terms throughout 4.1--it is challenging to understand precisely what are the main ingredients of Sparse-PGD, why they matter, and what decisions influenced their design.

2. Evaluation could be clearer - While I appreciate the extensive evaluation, it does not appear to disclose sufficient information to measure the performance of sAA. Specifically, (a) a distortion vs accuracy curve should be plotted, so that we can understand the performance curves of sAA against baselines. Reporting the final results at a fixed norm boundary is not readily indicative of attack performance, given that are are many values of k a defender would consider to be "adversarial", (b) when attacking against adversarially trained models, perturbations must stay within the threat model. That is, it should be made clear that, when attacking an l-infinity-based model, the l0 perturbations also do not exceed, e.g., 8/255. Otherwise, it is not clear to me what insights are to be drawn from attacking a model whose threat model is violated, (c) mixing threat models does not seem sound. It is not clear why black-box attacks are compared to white-box attacks, etc. White-box threat models should only be compared to white-box attacks, and likewise for black-box attacks.

3. Attack configuration does not seem fair - It is not clear to me why Sparse-RS is included within sAA when it is used verbatim from prior work. So that readers can understand the core contributions of this work, comparisons against baselines should only be evaluated against the introduced attacks. Moreover, the JSMA (Papernot, 2016) is one of the first l0-based attacks to be introduced in the literature. It is unclear to me why this not compared against in this work.

---

> ### Author Response · Authors · 2023-11-22
> **Rebuttal [1/3]**
>
> Dear reviewer DCtb,
>
> Thanks for your insightful and constructive comments. In response to your questions, we offer the following point-to-point answers:
>
> 1. For updating the magnitude tensor, are p and delta the same variable?
>
>     **Reply**: In our work, we decompose the sparse perturbation $\delta$ into a magnitude tensor $p\in\mathbb{R}^{h\times w\times c}$ and a sparsity mask $m\in\{0,1\}^{h\times w\times 1}$, i.e., $\delta =p \odot m$. We update $p$ and $m$ separately, and then obtain $\delta$ with the updated $p$ and $m$.
>
> 2. Why is the l2-norm of the loss taken in (5)?
>
>     **Reply**: According to the first-order Taylor expansion of loss function, $L(x+v)\approx L(x)+\nabla L(x)^T v$. Finding the steepest ascent direction of the loss function is equivalent to finding the $v$ which maximizes the directional derivative. In Euclidean space, if we want to find the steepest ascent direction within the step size of the unit norm, i.e., ||v||2=1 , the direction is $v=\nabla L(x)/||\nabla L(x)||_2$ [1].
> In our scenarios, we want to update $\widetilde{m}$ with the step size $\beta$, so the updating can be written as $\widetilde{m} = \widetilde{m} + \beta \cdot \nabla_{\widetilde{m}} L / (||\nabla_{\widetilde{m}} L||_2 + \gamma)$, where $\gamma$ is a small constant to prevent the denominator from being zero.
>
> 3. For updating the sparsity mask, what is gradient ascent performed on?
>
>     **Reply**: The $l_0$ bounded perturbation $\delta$ can be decomposed into magnitude tensor $p$ and a sparsity mask $m$, i.e., $\delta =p \odot m$. We aim to maximize the loss $L(\theta, x+p\odot m)$ through updating $p$ and $m$ with gradient ascent. Since $m$ is sparse and discrete, directly optimizing it is challenging. Instead, we introduce a continuous variable $\widetilde{m} \in \mathbb{R}^{h\times w\times 1}$ and project $\widetilde{m}$ to the feasible set $S_m$ before multiplying it with the magnitude tensor $p$ to calculate the sparse perturbation $\delta$. Since $\widetilde{m}$ is a continuous variable, we can optimize it using gradient ascent.
>
> 4. It is unclear what, "Since elements in m are 0 or 1, we use sigmoid to normalize elements in m-tilde to be 0 or 1" is trying to say; aren't elements in m in [0, 1] because of (6)?
>
>     **Reply**:  We are sorry about the confusion, we have modified the claim in the paper. In addition, we clarify that the sigmoid function used here is to prevent $\widetilde{m}$ from being too large in magnitude. When the magnitude of $\widetilde{m}$ is large, its gradient will vanish to zero by applying the sigmoid function, this mechanism will prevent $\widetilde{m}$ from being too large and ensure the numeric stability. In the same settings of Table 2 in section 5.3, if we discard the sigmoid function, the robust accuracy increases from 20.0 to 68.5, which further demonstrates the effectiveness of the sigmoid function.
>
> 5. The argument that projection on the binary set Sm is discarded because coordinate descent is suboptimal is unclear; why is such a projection introduced to be later argued as suboptimal and thus discarded? (In fact, this observation is stated twice)
>
>     **Reply**: Due to $S_{m}$, the gradient of $p$ is sparse, so at most $k$ elements of $p$ are updated at one iteration, which is equivalent to $k$-coordinate descent. Previous work [2] already demonstrates that coordinate descent results in slow convergence and suboptimal performance. In addition, we want to emphasize that the projection operator is only discarded during the back-propagation, it still works in the forward propagation. Moreover, as we indicate in Table 3, the projected and unprojected gradients are complementary in terms of performance, none of them is always better than the other, so we run sPGD twice with different backward functions in sAA for comprehensive evaluation.
>
> 6. It is unclear where the projection onto the binary set Sm is used in gp and why it is used in tandem with gp-tilde if gp exhibits both non-convergence and gradient explosions
>
>     **Reply**: This question is already addressed in the explanations above.

---

> ### Author Response · Authors · 2023-11-22
> **Rebuttal [2/3]**
>
> 7. There are many terms that are co-dependent with other terms throughout 4.1--it is challenging to understand precisely what are the main ingredients of Sparse-PGD, why they matter, and what decisions influenced their design.
>
>     **Reply**: The main idea of our method is to decompose a $l_0$ bounded perturbation $\delta$ into a magnitude tensor $p$ and a sparsity mask $m$.
>
>     - Since $p$ is only constrained in the image box $[0, 1]^{h\times w\times c}$ which is similar to the $l_\infty$ constraint, we update it using $l_\infty$-PGD.
>     - Updating a sparse and discrete $m$ is challenging in optimization, we update its continuous alternative $\widetilde{m}$ and then projecting $\widetilde{m}$ to obtain $m$.
>     - Due to the binary projection onto $S_{m}$, the update of $p$ is equivalent to $k$-coordinate descent, which leads to slow convergence and suboptimal performance. Inspired by training pruned neural networks and lottery ticket hypothesis [3-6], we discard the projection $S_{m}$ when calculating the gradient of $p$ to ensure every element of $p$ can be updated. However, in the experiments, we found the unprojected and projected gradients are complementary in terms of performance.
>     - Random reinitialization is proposed to prevent $m$ from getting trapped into local maxima, because $m$ changes only when the relative magnitude ordering of the continuous alternative $\widetilde{m}$ changes. In other words, slight changes in $\widetilde{m}$ usually mean no change in $m$.
>
> 8. A distortion vs accuracy curve should be plotted, so that we can understand the performance curves of sAA against baselines. Reporting the final results at a fixed norm boundary is not readily indicative of attack performance, given that are are many values of k a defender would consider to be "adversarial"
>
>     **Reply**: Thank you for pointing this out, we added the curves of sAA in Figure 1 (b).
>
> 9. When attacking against adversarially trained models, perturbations must stay within the threat model. That is, it should be made clear that, when attacking an l-infinity-based model, the l0 perturbations also do not exceed, e.g., 8/255. Otherwise, it is not clear to me what insights are to be drawn from attacking a model whose threat model is violated
>
>     **Reply**: In the previous work [7], when evaluating the robustness of an $l_\infty$-robust model against $l_2$ attacks, they do not constrain the $l_\infty$ norm of the $l_2$ perturbations to be $8/255$ or something else. The results in [7] also indicate that a model that is robust to a specific attack is usually not robust to another attack at all. In addition, when we validate whether a model is robust to a white-box attack, we should follow the principle that the attacker knows model architecture and model weights; training algorithm and training data; test time randomness (either the values chosen or the distribution) of the defender [8]. In the evaluation, we should not compromise the defender, so what we do is just to generate $l_0$ bounded perturbations without any other restrictions.
>
> 10. Mixing threat models does not seem sound. It is not clear why black-box attacks are compared to white-box attacks, etc. White-box threat models should only be compared to white-box attacks, and likewise for black-box attacks.
>
>     **Reply**: As shown in AutoAttack [9], they also combine white-box and black-box attacks to obtain a reliable and comprehensive evaluation of adversarial robustness, since white-box attacks sometimes have suboptimal performance due to gradient masking, but black-box attacks are immune from it because they do not rely on the gradient information. Empirically, among current adversarial attacks generating $l_0$ bounded perturbations, black-box attacks usually outperform white-box ones. For example, we see black-box sparse-RS outperforms white-box PGD$_0$ in many cases, depending on the evaluated models. In summary, we need to combine both white-box and black-box attacks to reliably and comprehensively evaluate the robustness, because they are complimentary.
>
> 11. Attack configuration does not seem fair - It is not clear to me why Sparse-RS is included within sAA when it is used verbatim from prior work. So that readers can understand the core contributions of this work, comparisons against baselines should only be evaluated against the introduced attacks. Moreover, the JSMA (Papernot, 2016) is one of the first l0-based attacks to be introduced in the literature. It is unclear to me why this not compared against in this work.
>
>     **Reply**: Thanks for pointing this out, we have cited it in the paper. Although JSMA is one of the first $l_0$ attacks, its code is not publicly available, making it difficult for reproduction. In addition, JSMA is not included in the existing works we use as baselines either. Considering these two points, we do not include it in the comparison.

---

> > ### Author Response · Authors · 2023-11-22
> > **Rebuttal [3/3]**
> >
> > > [1] Boyd, S., \& Vandenberghe, L. (2004). Convex optimization. Cambridge university press.
> >
> > > [2] Yulun Jiang, Chen Liu, Zhichao Huang, Mathieu Salzmann, and Sabine Süsstrunk. Towards stable and efficient adversarial training against l1 bounded adversarial attacks. In International Conference on Machine Learning. PMLR, 2023.
> >
> > > [3] Jonathan Frankle and Michael Carbin. The lottery ticket hypothesis: Finding sparse, trainable neural networks. In International Conference on Learning Representations, 2019.
> >
> > > [4] Vivek Ramanujan, Mitchell Wortsman, Aniruddha Kembhavi, Ali Farhadi, and Mohammad Rastegari. What’s hidden in a randomly weighted neural network? In Proceedings of the IEEE/CVF
> > conference on computer vision and pattern recognition, pp. 11893–11902, 2020.
> >
> > > [5] Yonggan Fu, Qixuan Yu, Yang Zhang, Shang Wu, Xu Ouyang, David Cox, and Yingyan Lin. Drawing robust scratch tickets: Subnetworks with inborn robustness are found within randomly initialized networks. Advances in Neural Information Processing Systems, 34:13059–13072, 2021.
> >
> > > [6] Chen Liu, Ziqi Zhao, Sabine Süsstrunk, and Mathieu Salzmann. Robust binary models by pruning randomly-initialized networks. Advances in Neural Information Processing Systems, 35:492–506, 2022.
> >
> > > [7] Francesco Croce, Matthias Hein. Adversarial Robustness against Multiple and Single-Threat Models via Quick Fine-Tuning of Robust Classifiers. ICML 2022.
> >
> > > [8] Athalye, Anish, Nicholas Carlini, and David Wagner. "Obfuscated gradients give a false sense of security: Circumventing defenses to adversarial examples." International conference on machine learning. PMLR, 2018.
> >
> > > [9] Francesco Croce and Matthias Hein. Reliable evaluation of adversarial robustness with an ensemble of diverse parameter-free attacks. In International conference on machine learning, pp. 2206–2216. PMLR, 2020

---

### Official Review · Reviewer_DbyG · 2023-11-01

**Soundness:** 2 fair
**Presentation:** 3 good
**Contribution:** 2 fair
**Rating:** 3
**Confidence:** 5

**Summary:**

The paper proposes a variant of PGD for $\ell_0$-bounded adversarial perturbations, named Sparse-PGD (sPGD), which jointly optimizes a dense perturbation and a sparsity mask. Then, sPGD, on different loss functions and with two alternative formulations, is used to form, together with an existing black-box attack, Sparse-AutoAttack (sAA), which aims at extending the AutoAttack to the $\ell_0$-threat model. In the experiments on CIFAR-10 and CIFAR-100, leveraging its multiple components, sAA improves upon the robustness evaluation of existing attacks. Finally, sPGD is used in adversarial training to achieve SOTA $\ell_0$-robustness.

**Strengths:**

- Adapting PGD to optimize $\ell_0$-bounded attacks is a challenging task, and sPGD is shown to often outperform existing attacks, especially white-box ones. Moreover it can be integrated into the adversarial training framework.

- Extending AA to the $\ell_0$-threat model would be important, and sPGD might be a promising step in such direction.

**Weaknesses:**

- While sAA seems effective (Table 1), there are some concerns in my opinion: first, according to Fig. 1a, the attacks notably benefit from more iterations. In particular, Sparse-RS shows significant improvements between 3k and 10k iterations for all models, which means that the results reported in Table 1 might be suboptimal. Second, in Tables 6, 7 and 8, CS alone appears to be better than sAA on the models robust to $\ell_0$-attacks: while CS is evaluated on a subset of points only, an improvement of more than 3% (Table 6) seems significant to hint to the fact that, even on the full test set, the results of sAA might be improved. Finally, in most cases the robust accuracy of the best individual attack (either RS or sPGD) is quite higher (2-3%) than their worst-case, i.e. sAA, which suggests that each attack is suboptimal.

- The budget of iterations of the attacks is not justified: looking at Fig. 1a it seems that more iterations would significantly improve the results, especially for RS. If I understand it correctly, sPGD is used for 20 runs ({1 CE, 9 targeted CE} x {projected, unprojected}) each of 300 iterations, for total 6k iterations (each consisting in one forward and one backward pass of the network). However, only 3k queries (forward pass only) are used for RS, which seems unbalanced given that RS provides better results for $\ell_\infty$- and (especially) $\ell_0$-adversarially trained models.

- The claim that no prior works proposed adversarial training for the $\ell_0$-threat model is imprecise, see e.g. [Croce & Hein (2019)](https://openaccess.thecvf.com/content_ICCV_2019/papers/Croce_Sparse_and_Imperceivable_Adversarial_Attacks_ICCV_2019_paper.pdf). Moreover, the cost of using 100 iterations of sPGD in adversarial training seem very large. Finally, the sAT and sTRADES would need to be added to Fig. 1a, to see how the effect of more queries in RS on the achieved accuracy (see previous points).

**Questions:**

The main concerns are detailed above. As minor point, it would be interesting to have some evaluation on ImageNet models.

Overall, I like the idea of extending AA to the $\ell_0$-threat model, but the current results do not convincingly support how the paper proposes to build sAA (e.g. how significantly would the results improve with 2x iterations to every attack, of 4x to RS?). Similarly, the effectiveness of adversarial training with sPGD should be tested more thoroughly.

---

> ### Author Response · Authors · 2023-11-22
> **Rebuttal [1/2]**
>
> Dear reviewer DbyG,
>
> Thanks for your insightful and constructive comments. In response to your questions, we offer the following point-to-point answers.
>
> For problems mentioned in Weaknesses:
>
> 1. \>\>\> While sAA seems effective (Table 1), there are some concerns in my opinion: first, according to Fig. 1a, the attacks notably benefit from more iterations. In particular, Sparse-RS shows significant improvements between 3k and 10k iterations for all models, which means that the results reported in Table 1 might be suboptimal. Second, in Tables 6, 7 and 8, CS alone appears to be better than sAA on the models robust to $l_0$-attacks: while CS is evaluated on a subset of points only, an improvement of more than 3\% (Table 6) seems significant to hint to the fact that, even on the full test set, the results of sAA might be improved. Finally, in most cases the robust accuracy of the best individual attack (either RS or sPGD) is quite higher (2-3\%) than their worst-case, i.e. sAA, which suggests that each attack is suboptimal.
>
>     **Response**: First, Sparse-RS indeed has performance improvement between 3k and 10k iterations for all models. However, as illustrated in Figure 1 (a), in the cases where our method outperforms Sparse-RS at 3k iterations, it is still substantially better than Sparse-RS when we use more iterations. That is to say, setting the number of iterations to 3k does not affect the performance superiority of our method. However, we are happy to present the results of Sparse-RS with 10k iterations and compare them with our methods. Due to the time constraint, we only run the experiments on a subset of models as shown in Table 1. It is clear that sAA still performs the best in all models except in $\ell_{\infty}$-adv model and we acknowledge this as one weakness of our method and will try to improve it in future work. As for the second question, we would like to emphasize that our $l_0$-robust models are just baselines for future work. The major motivation of $l_0$-robust models is to demonstrate the efficiency of our method so that we can utilize it for adversarial training. Due to the challenges of optimization in an $l_0$ bounded space, we notice that the $l_0$-robust models are more vulnerable to black-box attacks, especially sparse-RS, although these models still obtain the best robustness when we consider all attack methods. In Tables 6, 7 and 8, the reason why CS is better than sAA on the models robust to $l_0$-attacks can be attributed to that Sparse-RS underperforms CS on $l_0$-robust models. Despite that, due to the very high complexity of CS (10x slower), we do not adopt it as the black-attack here. We believe this issue can be solved if we have a better adversarial training methods to mitigate the gradient masking problem of the models obtained, which we leave it as future work.
>
>     ***Table 1: Robust accuracy of selected models on Sparse RS where the sparsity level $k=20$. We record the performance of RS in both 3k and 10k iterations and their changes.***
>
>     |Model | Network | RS  | **sPGD$_{\mathrm{CE}}$** |**sPGD$_{\mathrm{CE+T}}$** | **sAA**|
>     |----|----|----|----|----|----|
>     |Vanilla | RN-18 | 0.0 $\to$ 0.0 | 0.0 |0.0 | **0.0**|
>     |$l_\infty$-adv. trained, $\epsilon=8/255$|
>     | PORT | RN-18 |14.6 $\to$ **9.7**|20.9|16.1 | 10.8|
>     |DKL| WRN-28 | 11.0 $\to$ **7.0** | 22.5|16.6| 9.6|
>     |$l_2$-adv. trained, $\epsilon=0.5$|
>     |HAT | PRN-18 |20.5 $\to$ 13.9 | 13.2 | 10.0 | **9.3**|
>     |DM| WRN-28 | 23.4 $\to$ 16.1 |20.9| 15.5| **14.1**|
>     |$l_1$-adv. trained, $\epsilon=12$|
>     |$l_1$-APGD | PRN-18 |33.1 $\to$ 25.8 |  24.0| 20.0|**19.5**|
>     |$l_0$-adv. trained, $k=20$|
>     |**sTRADES** | PRN-18 |52.1 $\to$ **41.8** |79.5|77.9 |52.0 |
>
> 2.  \>\>\> The budget of iterations of the attacks is not justified: looking at Fig. 1a it seems that more iterations would significantly improve the results, especially for RS. If I understand it correctly, sPGD is used for 20 runs ({1 CE, 9 targeted CE} x {projected, unprojected}) each of 300 iterations, for total 6k iterations (each consisting in one forward and one backward pass of the network). However, only 3k queries (forward pass only) are used for RS, which seems unbalanced given that RS provides better results for $l_0$- and (especially) $l_0$-adversarially trained models.
>
>     ***Response***: We need to clarify that the sPGD with 20 runs ({1 CE, 9 targeted CE} x {projected, unprojected}) is only adopted in sAA. We do not compare the performance of Sparse-RS with sPGD and its variants. By contrast, in Figure 1 (a), we evaluate sPGD with unprojected gradient with different numbers of iterations. The robust accuracies of Sparse-RS and sPGD at the same x-axis value are obtained with the same number of iterations. Therefore, we think the comparison is fair here.

---

> > ### Author Response · Authors · 2023-11-22
> > **Rebuttal [2/2]**
> >
> > 3. The claim that no prior works proposed adversarial training for the $l_0$-threat model is imprecise, see e.g. Croce \& Hein (2019). Moreover, the cost of using 100 iterations of sPGD in adversarial training seem very large. Finally, the sAT and sTRADES would need to be added to Fig. 1a, to see how the effect of more queries in RS on the achieved accuracy (see previous points).
> >
> >     **Response**: First of all, thank you for pointing this out, we have modified this claim and updated the results of using PGD$_0$ as adversarial training in the paper. Meanwhile, we have also included the results in Table 1 and General Response for your reference. It is clear that models adversarially trained by PGD$_0$ attack are not robust against our proposed attack and less robust than the model trained using our method. Secondly, we want to emphasize that using 100 iterations of sPGD in adversarial training is just a baseline to obtain $l_0$-robust model, we will conduct research on a more efficient adversarial training algorithm for $l_0$ bounded perturbations in future work.
> >
> > For the question mentioned in Questions:
> >
> > 1. Overall, I like the idea of extending AA to the $l_0$-threat model, but the current results do not convincingly support how the paper proposes to build sAA (e.g. how significantly would the results improve with 2x iterations to every attack, of 4x to RS?). Similarly, the effectiveness of adversarial training with sPGD should be tested more thoroughly.
> >
> >     **Response**: The results of ImageNet100 have been added to Table 9 of Appendix A.1 to more comprehensively demonstrate the effectiveness of our methods, especially in the case of high-resolution images. Similar to AutoAttack, which includes a standard version and a more expensive AutoAttack+ in its repository https://github.com/fra31/auto-attack, we will provide sAA of different levels, such as the standard one as demonstrated in the paper and the stronger ones with more iterations for each attack, to practitioners to evaluate the robustness of their models based on their computational constraints.
> >
> > > [1] Francesco Croce, Maksym Andriushchenko, Naman D Singh, Nicolas Flammarion, and Matthias Hein. Sparse-rs: a versatile framework for query-efficient sparse black-box adversarial attacks. In Proceedings of the AAAI Conference on Artificial Intelligence, volume 36, pp. 6437–6445, 2022.

---

> > > ### Comment · Reviewer_DbyG · 2023-11-22
> > >
> > > I thank the authors for the response and additional experiments.
> > >
> > > The new results show that Sparse-RS with 10k iterations is better than sAA (which even contains Sparse-RS with 3k iterations) in 3/6 cases (all attacks are equally good on the vanilla model). Also, the improvement from 3k to 10k iterations is significant on all classifiers, and it's not clear how the results would evolve with even more queries. The point is that it's not clear that sAA consistently provides better performance than the existing methods, and the budget given to each of the attacks is not sufficiently to achieve good performance. This is also supported by CS sometimes being more effective than sAA (I don't think that the fact that CS is computationally expensive and can't scale to large models is relevant, since it just shows that the evaluation of sAA might be suboptimal). Moreover, the largest drop in robust accuracy from using more iterations happens on the sTRADES model, and then it's not clear how effective the proposed method is for adversarial training.
> > >
> > > Overall, the rebuttal doesn't solve the main concerns mentioned in the initial review.

---

### Official Review · Reviewer_Ngmx · 2023-11-04

**Soundness:** 2 fair
**Presentation:** 3 good
**Contribution:** 2 fair
**Rating:** 5
**Confidence:** 4

**Summary:**

This paper proposed a ﻿﻿effective and efficient attack called ﻿sparse-PGD (sPGD) to generate sparse adversarial perturbations bounded by l_{0} norm, which achieves better performance with ﻿a small number of iterations. Sparse-AutoAttack (sAA) is presented﻿, which is the ensemble of the white-box sPGD and another black-box sparse attack, for reliable robustness evaluation against l_{0} bounded perturbations. ﻿Furthermore, adversarial training is conducted against l_{0} bounded sparse perturbations. The model trained with the proposed attack is superior to other ﻿sparse attacks regarding robustness.

**Strengths:**

+ The attacks are evaluated under different norms and limited iterations for fair comparison.
+ The white-box and black-box are combined ﻿for comprehensive robustness evaluation.
+ The impacts of ﻿﻿Iteration Number and Sparsity Level are considered and analyzed.

**Weaknesses:**

- Following ﻿Sparse Adversarial and Interpretable Attack Framework (SAIF) [1], which adopts ﻿a magnitude tensor and sparsity mask same as this paper, the authors further ﻿discard the projection to the binary set when calculating the gradient and use the unprojected gradient to update ﻿the magnitude tensor p. ﻿Sparse-AutoAttack (sAA) part has extended the work of ﻿AutoAttack (AA) [2,3], and the reason for discarding ﻿the adaptive step size, momentum and difference of logits ratio (DLR) loss function should be further explained clearly. The paper appears to offer limited new perspectives on the attack process and lacks a notable degree of technical innovation.
- The authors claim that “﻿We are the first to conduct adversarial training against l_{0} bounded perturbations.” However, related work had also conducted similar experiments [4].
- This paper has emphasized the contribution of ﻿computational complexity and efficiency but lacks corresponding analysis for ﻿computational complexity and query budgets for comparison.
- In Table 1 in experimental part, ﻿RS attack outperforms sPGD_{CE+T} for l_{∞} models while more analysis is required.
- The performance analysis in Subsection 5.1 is not well-organized for clarity.
- Many parameters in this paper need to be pre-defined. For example, ‘the current sparsity mask remains unchanged for three consecutive iterations, the continuous alternative fm will be randomly reinitialized for better exploration. ‘ Why three consecutive iterations? Will choosing a different number affect the results?  What is \alpha and \beta? Will \alpha and \beta affect the value of ‘three iterations’? Also for a small \lambda, it is unclear about how small the \lambda should be.
- How do you set up the budget for each attack method to compute the robust accuracy so the comparison is fair?

References

[1] ﻿Tooba Imtiaz, Morgan Kohler, Jared Miller, Zifeng Wang, Mario Sznaier, Octavia Camps, and Jennifer Dy. Saif: Sparse adversarial and interpretable attack framework. arXiv preprint arXiv:2212.07495, 2022.
[2] ﻿Francesco Croce and Matthias Hein. Reliable evaluation of adversarial robustness with an en- semble of diverse parameter-free attacks. In International conference on machine learning, pp. 2206–2216. PMLR, 2020.
[3] ﻿Francesco Croce and Matthias Hein. Mind the box: l_1-apgd for sparse adversarial attacks on image classifiers. In International Conference on Machine Learning, pp. 2201–2211. PMLR, 2021.
[4] ﻿Francesco Croce and Matthias Hein. Sparse and imperceivable adversarial attacks. In Proceedings of the IEEE/CVF international conference on computer vision. 2019

**Questions:**

Pls see the Section Weaknesses

---

> ### Author Response · Authors · 2023-11-22
> **Rebuttal [1/3]**
>
> Dear reviewer Ngmx,
>
> Thanks for your insightful and constructive comments. In response to your questions, we offer the following point-to-point answers:
>
> 1. \>\>\> Following Sparse Adversarial and Interpretable Attack Framework (SAIF) [1], which adopts a magnitude tensor and sparsity mask same as this paper, the authors further discard the projection to the binary set when calculating the gradient and use the unprojected gradient to update the magnitude tensor p. Sparse-AutoAttack (sAA) part has extended the work of AutoAttack (AA) [2,3], and the reason for discarding the adaptive step size, momentum and difference of logits ratio (DLR) loss function should be further explained clearly. The paper appears to offer limited new perspectives on the attack process and lacks a notable degree of technical innovation.
>
>     **Response**:  For the attack process, though we adopt a similar decomposition method to the perturbation to SAIF [1], **our attack design is significantly different from the previous work in two perspectives:**
>     1. Instead of using an $\ell_1$ convex surrogate for the $l_0$ norm constraint of the sparsity mask as done in SAIF, we introduce a continuous mask $\widetilde{m}$ and project it back to the feasible set. To prevent the magnitude of $\widetilde{m}$ from being too big, we add a sigmoid function before projecting $\widetilde{m}$ to the binary mask $m$. This **new but simple** strategy has proven to be effective as shown in the ablation studies (Table 2, Sec 5.3) where we observed a huge robust accuracy drop from 49.5\% to 22.2\%. Moreover, we further improve the exploration capability for the mask $m$ by a random reinitialization mechanism, which is shown to improve the performance in the ablation study. This design alleviates the problem of local convergence, which is reported in SAIF.
>     2. The ablation studies (Table 2, Sec 5.3) also indicated naively decomposing the perturbation by $\delta = p \odot m$ can **deteriorate the performance**, which further corroborates the effectiveness of our attack designs.
>
>     In terms of backward function for updating $p$, we argue that the original gradient $g_{p} = \nabla_{\delta} L(\theta, x + \delta) \odot m$ is sparse due to the sparsity of the mask $m$. **The sparse update will result in sub-optimal performance** similar to coordinate descent when there are limited iterations. To solve this problem, we adopt a novel unprojected gradient $g_{p} = \nabla_{\delta} L(\theta, x + \delta) \odot \sigma(\widetilde{m})$ inspired by network pruning and lottery ticket hypothesis [6 - 9] and combine it with the original sparse gradient to achieve a better trade-off on exploration and exploitation.
>
> 2. \>\>\> The authors claim that “We are the first to conduct adversarial training against $l_{0}$ bounded perturbations.” However, related work had also conducted similar experiments [4].
>
>    **Response**: Thank you for pointing this out. We will modify this claim in the paper and add the results of using PGD$_0$ as adversarial training. The results are reported in Table 1, Sec 1.2.1 of General Response, and we have also included them in the paper. We observe that the model trained with adversarial samples generated by PGD$_0$ can be easily beaten by our proposed attack sPGD, while the model trained with sPGD-generated adversarial samples is still robust against PGD$_0$, so we do not include it in other experiments.
>
> 3. \>\>\> This paper has emphasized the contribution of computational complexity and efficiency but lacks corresponding analysis for computational complexity and query budgets for comparison.
>
>     **Response**:  In Section 5.2, we study the efficiency by comparing the performance of Sparse-RS with sPGD_CE (Figure 1(a)) and sPGD_CE+T (Figure 2) under different number of iterations. Both figures show that sPGD achieves significantly better performance than RS attack under limited iterations. That is to say, Sparse-RS requires a much larger query budget to achieve comparable performance.
>
>
> 4. \>\>\> In Table 1 in experimental part, RS attack outperforms sPGD_CE+T for $l_{\infty}$ models while more analysis is required.
>
>     **Response**: Thanks for pointing this out. For attacking $l_\infty$ models, we agree that sPGD is outperformed by RS when there are enough iterations, which means sPGD suffers from gradient masking to some degree in this case. This is also why we included the black-box RS attack in sAA to comprehensively evaluate the robustness. Further mitigating the gradient masking effect when sPGD is applied to some specific models will be our future work.

---

> > ### Author Response · Authors · 2023-11-22
> > **Rebuttal [2/3]**
> >
> > 5. \>\>\> The performance analysis in Subsection 5.1 is not well-organized for clarity.
> >
> >     **Response**: Sorry for the obscurity. We summarized the key points of the analysis below:
> >     - By comparing against five attacks, our proposed method archives the lowest robust accuracy among white-box attacks.
> >     - For black-box attacks, sPGD outperforms CS attack in terms of efficiency and performs better than RS attack in $\ell_1 \text{ and } \ell_2 $ models. Meanwhile, we acknowledge that sPGD's performance in $\ell_{\infty}$ models is weaker than RS attack. We argue that this may result from the existence of gradient masking. As future work, we will try to investigate the causes and improve the algorithm.
> >     - Lastly, we trained the PRN-18 model against our attack in two fashions: vanilla adversarial training (sAT) and TRADES (sTRADES). We showed that models trained by these two methods against sPGD yield the strongest robustness compared with other models including PGD$_0$-A and PGD$_0$-T. We consider these two models as strong baselines for $\ell_0$-adv models and defer a more thorough investigation on robust learning using sPGD as future work.
> >
> > 6. \>\>\> Many parameters in this paper need to be pre-defined. For example, ‘the current sparsity mask remains unchanged for three consecutive iterations, the continuous alternative fm will be randomly reinitialized for better exploration. ‘ Why three consecutive iterations? Will choosing a different number affect the results? What is $\alpha$ and $\beta$? Will $\alpha$ and $\beta$ affect the value of ‘three iterations’? Also for a small $\lambda$, it is unclear about how small the $\lambda$ should be.
> >
> >     \textbf{Response: } When designing the algorithm, we follow the philosophy of AutoAttack to hard-code some hyper-parameters.
> >     This design is motivated by the observation that the model performance is not sensitive to these hyper-parameters, it facilitates the practitioners to avoid tuning additional hyper-parameters.
> >
> >     Empirically, we found "Three consecutive iterations" works quite well. In addition, we conduct an ablation study on different reinitialization schemes. The result shown below \textbf{indicates that our method is quite robust to this hyperparameters.}
> >
> >     $\alpha$ and $\beta$ are the step sizes for updating magnitude tensor $p$ and continuous mask alternative $\widetilde{m}$, which are set to $1/4$ and $1/4 \times \sqrt{h \times w}$, respectively.
> >     Here, $h$ and $w$ are the height and width of the input image $x$ (both $h$ and $w$ are $32$ in CIFAR-10 dataset).
> >     In fact we have tested different values of these two hyper-parameters ($\alpha \in \{\frac{1}{16}, \frac{1}{8}, \frac{1}{4}, \frac{1}{2}, \frac{3}{4}, 1\}, \beta \in \{2,4,8,16,24,32\}$)，the results are included in Appendix A.3 (Table 9 and 10).
> >     The results show that \textbf{the performance is not sensitive to the change of the two hyper-parameters.}
> >
> >     For the question regarding $\lambda$, we assume you were referring to the parameter $\gamma$ when updating the gradient of $\widetilde{m}$.
> >     In fact, it is common practices to use a small constant to avoid zero division. As a result, the choice of gamma will not greatly affect the performances. In our experiments, we set it to be $1e-10$.
> >
> > 7. \>\>\> How do you set up the budget for each attack method to compute the robust accuracy so the comparison is fair?
> >
> >     **Response**:  To ensure a fair comparison, we set the budget for each attack method we compare with to 3k. For targeted attacks such as sPGD_CE+T, we need $10$ runs for different targets, so we run $300$ iterations for each target. As for sAA, following the setting of AutoAttack [5], we also run each sPGD_CE+T (unprojected and projected) and Sparse-RS with 3k iterations, respectively.
> >
> > ***Table 1: Ablation study on the patience for reinitialization.***
> >
> > |Reinitialization patience | 1 | 3 | 5 | 7 | 9|
> > |----|----|----|----|----|----|
> > Robust Accuracy | 20.2 | 20.0 | 20.1 |20.4 | 20.0|

---

> ### Author Response · Authors · 2023-11-22
> **Rebuttal [3/3]**
>
> References:
> > [1] Tooba Imtiaz, Morgan Kohler, Jared Miller, Zifeng Wang, Mario Sznaier, Octavia Camps, and Jennifer Dy. Saif: Sparse adversarial and interpretable attack framework. arXiv preprint arXiv:2212.07495, 2022.
>
> > [2] Francesco Croce and Matthias Hein. Reliable evaluation of adversarial robustness with an ensemble of diverse parameter-free attacks. In International conference on machine learning, pp. 2206–2216. PMLR, 2020.
>
> >[3] Francesco Croce and Matthias Hein. Mind the box: $l_1$-apgd for sparse adversarial attacks on image classifiers. In International Conference on Machine Learning, pp. 2201–2211. PMLR, 2021.
>
> >[4] Francesco Croce and Matthias Hein. Sparse and imperceivable adversarial attacks. In Proceedings of the IEEE/CVF international conference on computer vision. 2019.
>
> >[5] Croce, Francesco, and Matthias Hein. "Reliable evaluation of adversarial robustness with an ensemble of diverse parameter-free attacks." International conference on machine learning. PMLR, 2020.
>
> >[6] Jonathan Frankle and Michael Carbin. The lottery ticket hypothesis: Finding sparse, trainable neural networks. In International Conference on Learning Representations, 2019.
>
> >[7] Vivek Ramanujan, Mitchell Wortsman, Aniruddha Kembhavi, Ali Farhadi, and Mohammad Rastegari. What’s hidden in a randomly weighted neural network? In Proceedings of the IEEE/CVF conference on computer vision and pattern recognition, pages 11893–11902, 2020.
>
> >[8] Yonggan Fu, Qixuan Yu, Yang Zhang, Shang Wu, Xu Ouyang, David Cox, and Yingyan Lin. Drawing robust scratch tickets: Subnetworks with inborn robustness are found within randomly initialized networks. Advances in Neural Information Processing Systems, 34:13059–13072, 2021.
>
> >[9] Chen Liu, Ziqi Zhao, Sabine Süsstrunk, and Mathieu Salzmann. Robust binary models by pruning randomly-initialized networks. Advances in Neural Information Processing Systems, 35:492–506, 2022

---

### Author Response · Authors · 2023-11-22
**General Response**

We appreciate all reviewers' constructive and insightful comments and their hard work. First, we would like to list the revisions to the manuscript as follows, which are also highlighted in blue in the updated paper.

1. Modify the claim about adversarial training, and add the results and analysis of the models trained with PGD0 in Table of Sec. 5.1.
2. Modify the statement about sigmoid function in Sec. 4.1.
3. Add the curve of sAA in Figure 1 (b).
3. Add the results on ImageNet100 in Table 9 of Appendix A.1.

In addition, we make a general response to the question mentioned by more than one reviewer:
1. \>\>\> The claim that no prior works proposed adversarial training for the l0-threat model is imprecise, see e.g. Croce & Hein (2019).

    **Reply**: We apologize for failing to include the adversarially trained model of PGD0 [1] in the paper and hereby provide additional experiment results. Similar to sAT and sTRADES, we trained two models against PGD0 using vanilla adversarial training (PGD0-A) and TRADES (PGD0-T). Meanwhile, we have also included the results below and in Table 1. It is clear that models adversarially trained by PGD$_0$ attack are not robust against our proposed attack and less robust than the model trained using our method.

***Table 1: Robust accuracy of adversarially trained models on different attacks that generate $l_0$ bounded perturbations, where the sparsity level $k=20$.***
| Model | Network | Clean | CS | RS  | SF | PGD$_{0}$(CE+T) | SAIF (CE+T) |  **sPGD$_{\mathbf{\mathrm{CE}}}$** | **sPGD$_{\mathbf{\mathrm{CE+T}}}$**  | **sAA**|
|----|----|----|----|----|----|----|----|----|----|----|
| PGD$_0$-A | PRN-18 | 76.2 | 1.3 | 0.1| 17.1 | 0.0 | 1.3 | 0.0 | 0.0 | **0.0**|
| PGD$_0$-T | PRN-18 | 78.2 | 0.7 | 0.1|  16.6 | 0.0 | 0.7 | 0.0 | 0.0 | **0.0**|
|**sAT**| PRN-18 | 85.8| 48.1 | 45.1| 85.2 | 79.7 | 77.1|  78.2 | 76.8  |**44.6**|
|**sTRADES** | PRN-18 | 87.2 | 55.0 |52.1 |86.3|  82.2|79.2 |79.5|77.9 |**52.0**|

2.  Unclear difference with SAIF [2]

    **Reply**: For the attack process, though we adopt a similar decomposition method to the perturbation to SAIF=, **our attack design is significantly different from the previous work in two perspectives:**
    1. Instead of using an $\ell_1$ convex surrogate for the $l_0$ norm constraint of the sparsity mask as done in SAIF, we introduce a continuous mask $\widetilde{m}$ and project it back to the feasible set. To prevent the magnitude of $\widetilde{m}$ from being too big, we add a sigmoid function before projecting $\widetilde{m}$ to the binary mask $m$. This **new but simple** strategy has proven to be effective as shown in the ablation studies (Table 2, Sec 5.3) where we observed a huge robust accuracy drop from 49.5\% to 22.2\%. Moreover, we further improve the exploration capability for the mask $m$ by a random reinitialization mechanism, which is shown to improve the performance in the ablation study. This design alleviates the problem of local convergence, which is reported in SAIF.
    2. The ablation studies (Table 2, Sec 5.3) also indicated naively decomposing the perturbation by $\delta = p \odot m$ can **deteriorate the performance**, which further corroborates the effectiveness of our attack designs.

> [1] Francesco Croce and Matthias Hein. Sparse and imperceivable adversarial attacks. In Proceedings of the IEEE/CVF international conference on computer vision, pages 4724–4732, 2019.

> [2] Tooba Imtiaz, Morgan Kohler, Jared Miller, Zifeng Wang, Mario Sznaier, Octavia Camps, and Jennifer Dy. Saif: Sparse adversarial and interpretable attack framework. arXiv preprint arXiv:2212.07495, 2022.

---

### Meta-Review · Area_Chair_9ETf · 2023-12-05

**Metareview:**

This work focuses on sparse adversarial perturbations bounded by $l_0$ norm. They first propose a new white-box PGD-like attack named sparse PGD then propose a l0 autoattack based on several sparse attacks.

Strengths:

1. They propose an Autoattack version for sparse adversarial attacks, which I think is quite interesting.

Weaknesses:

1. The methods seem like a straightforward simplification for SAIF and lack proof that such optimization can get the optimal solution.

2. Its performance is also worse than Sparse-RS. Furthermore, they lack enough evaluation to demonstrate the efficiency they claimed since I think comparing the first convergence speed for a white box method and a black box method (Sparse RS) is unfair, not to mention Sparse-RS seems to perform better when the iteration number is large.

3. As for sAA, it seems that the main efforts of the attack is caused by Sparse-RS, which made such a contribution less convincing.

Therefore, I agree with the reviewers and tend to reject this paper.

**Justification For Why Not Higher Score:**

Their claims are not well proved and their contributions seem limited.

**Justification For Why Not Lower Score:**

N/A

---

### Decision · Program_Chairs · 2024-01-16

Reject